# **Cloud-top microphysics evolution in the Gamma phase space from a modeling perspective**

Lianet H. Pardo<sup>1</sup>, Luiz A. T. Machado<sup>1</sup>, and Micael A. Cecchini<sup>2</sup>

<sup>1</sup>Centro de Previsão de Tempo e Estudos Climáticos, Instituto Nacional de Pesquisas Espaciais, Cachoeira Paulista, Brasil <sup>2</sup>Departamento de Ciências Atmosféricas, Instituto de Astronomia, Geofísica e Ciências Atmosféricas, Universidade de São Paulo, Brasil

Correspondence to: Lianet H. Pardo (lianet.pardo@cptec.inpe.br)

**Abstract.** This research employs the recently introduced Gamma phase space to study the evolution of warm cloud microphysics, to evaluate different microphysics parameterizations and to propose an adjustment to bulk schemes for an improved description of cloud droplet size distributions (DSDs). A bin parameterization is employed to describe the main features of observed cloud-top DSD paths in the Gamma phase space. The modeled DSD evolution during the warm cloud life cycle is

- compared to the results obtained from HALO airplane measurements during the ACRIDICON-CHUVA campaign in the Amazon dry-to-wet season transition. The comparison shows an agreement between the observed and simulated trajectories in the Gamma phase space, providing a suitable qualitative representation of the DSD evolution. The degree of similarity between the trajectories is defined by the conditions of the environment, such as the aerosol number concentration, which modify the DSD evolution through modulation of its driving forces. The modeled DSD properties were also projected in the  $N_d - D_{eff}$  space
- to obtain further insights into their life cycle. Two different bulk microphysics parameterizations were evaluated regarding the evolution of the DSD and using the bin scheme as a reference. The results show the weakness of bulk schemes in representing trajectories in the Gamma phase space; thus, a new closure is proposed for better comparisons to the reference. The new closure resulted in an improvement in the representation of the DSD evolution, cloud droplet effective diameter and rain mixing ratio.

#### 1 Introduction

- Cloud microphysics parameterizations have strongly evolved, and new sets of schemes have been proposed over the past years (Khain et al., 2004; Gilmore et al., 2004; Khain et al., 2010; Mansell et al., 2010; Lim and Hong, 2010; Loftus et al., 2014; Thompson and Eidhammer, 2014). However, due to the complexities of the physical processes in determining the evolution of hydrometeor size distributions during the cloud life cycle, large uncertainties remain in all types of schemes. The lack of knowledge about the characteristics of the effects of atmospheric aerosols on clouds and precipitation is an important source
- of uncertainty in parameterizations, as are the descriptions of ice and mixed phase processes and the effects of turbulence and entrainment (Khain et al., 2015).

Although bin schemes are more accurate and flexible, their high computational cost makes them less useful for practical applications. Thus, bulk schemes are generally employed. However, the assumption of a predefined function for hydrometeor size distributions limits the range of situations that can be simulated with a reasonable degree of accuracy.

In bulk microphysics parameterizations, the gamma function is one of the more common ways to represent the droplet size distributions (DSDs) (Khain et al., 2015):

$$N(D) = N_0 D^{\mu} exp(-\Lambda D) \tag{1}$$

where  $N_0$  (cm<sup>-3</sup> $\mu$ m<sup>-1- $\mu$ </sup>),  $\mu$  (dimensionless) and  $\Lambda$  ( $\mu$ m<sup>-1</sup>) are the intercept, shape and curvature parameters, respectively, and N(D) is the number of droplets with diameter D per cm<sup>-3</sup> of air.

Most bulk microphysical parameterizations do not predict enough moments of hydrometeor size distributions to properly describe their variability. As a closure, the  $\mu$  parameter is commonly fixed or evaluated (Grabowski, 1998; Rotstayn and Liu, 2003; Morrison and Grabowski, 2007). Due to its functional relationship, the choice of  $\mu$  determines the values of  $N_0$  and  $\Lambda$ . Nevertheless, switching between different methods for  $\mu$  can lead to a 25% increase in cloud water path (Morrison

and Grabowski, 2007) and a 50% variation in the condensation rate (Igel and van den Heever, 2017). Thus, the description of hydrometeor size distributions in bulk parameterizations continues to be one of the major questions for the microphysics modeling community.

Cecchini et al. (2017b) introduced the phase space constituted by the three parameters ( $N_0$ ,  $\mu$  and  $\Lambda$ ) of the gamma function (Gamma phase space) as a very useful tool for tracking cloud microphysics evolution. They describe the possibility of inter-

15 preting microphysics processes as "pseudo-forces" in the Gamma phase space and show that different types of clouds have different trajectories in it, resulting from differences in the balance of those pseudo-forces, where the aerosol effect appears to play a major role. Their research provides a way to characterize the evolution of a cloud, opening new opportunities to understand cloud processes and to improve microphysics parameterizations.

The main goal of this study is to employ the Gamma phase space to evaluate the DSD evolution during the life cycle of warm clouds, as simulated by a bin microphysics scheme, and compare it with in-cloud measurement data provided by Cecchini et al. (2017b). The bin microphysics is used as the benchmark to study the performance of bulk parameterizations. Gamma DSDs are fitted to the bin model results to serve as the benchmark regarding the Gamma phase space. Furthermore, the pseudo-forces governing the displacements in this phase space are analyzed. According to the insights obtained from the bin simulations, a new approach to parameterize the  $\mu$  parameter in bulk schemes is proposed and tested.

# 25 2 Modeling approach

Warm microphysical processes were simulated using a bin parameterization (Tzivion et al., 1987; Feingold et al., 1988; Tzivion et al., 1989) inside a single-column model (Shipway and Hill, 2012), where the vertical velocity is prescribed.

Although we used the Kinematic Driver (KiD) as the single-column model, it is also designed to work with two spatial dimensions according to the objectives of the user. The prognostic variables are potential temperature (K) and water vapor,

30 hydrometeor and aerosol mixing ratios (kg kg<sup>-1</sup>). It uses the Exner pressure as a fixed vertical coordinate and the total variancediminishing scheme (Leonard et al., 1993) as the default advection scheme. Its prognostic variables are held on "full" model levels, while the vertical velocity and density are held on both "full" and "half" levels such that the grid can be used as a Lorenz-type (Lorenz, 1960) or Charney-Phillips-type (Charney and Phillips, 1953) grid.

The KiD model was conceived as a kinematic framework to compare different microphysics parameterizations without addressing the microphysics-dynamics feedbacks. Thus, obtaining precise quantitative simulations with KiD cannot be expected; nevertheless, it can provide important qualitative information about the behavior of hydrometeors during the life cycle of clouds.

In our simulations, a 1 s time step was used for both dynamics and microphysics algorithms during an integration time of 1200 s (20 min). For the vertical domain, a 120-level grid was defined with a 50-m grid spacing from 0 m to 6000 m of altitude. As initial conditions, vertical profiles of potential temperature and water vapor mixing ratio from an in situ atmospheric

sounding<sup>1</sup> were provided (Fig. 1a). We used the 12Z sounding from Boa Vista-RR, Brazil, for coherence with the atmospheric conditions where the data of the AC09 flight were collected. The potential temperature and water vapor profiles from that

sounding resembled the data measured by the AC09 flight, but with a greater resolution and vertical domain, thus making them more convenient to define the model initial conditions. The sounding data were interpolated to match the model resolution and then smoothed to represent a more general situation.

Here, the vertical velocity field (w(z,t)) was constructed based on the idea of having a layer of positive buoyancy, where a parcel updraft velocity would increase with height until reaching the negative buoyancy layer. The defined time dependence for the velocity maximum and its height roughly simulate the acceleration that the air must experience and the progressive

for the velocity maximum and its height roughly simulate the acceleration that the air must experience and the progressive destabilization of the air column (Fig. 1b).

$$w(z,t) = \begin{cases} W \sin\left(\frac{\pi}{2} \frac{t}{T}\right) e^{-\frac{1}{2}\log^2\left(0.004t - 0.0008z\right)} & (0.2z - t) < 0\\ 0 & otherwise \end{cases}$$
(2)

In Eq. 2, W represents the maximum updraft speed (with respect to both height and time) in m s<sup>-1</sup> and T is the length of the simulation in s. The value of W was set to 5 m s<sup>-1</sup> taking into account the measurements of the ACRIDICON-CHUVA AC09 flight (Wendisch et al., 2016), where the vertical velocity oscillated between 0 m s<sup>-1</sup> and 8 m s<sup>-1</sup> (Cecchini et al., 2017a).

#### 2.1 Microphysics representation

For the most realistic simulations performed in this work, we have used the  $TAU^2$  size-bin-resolved microphysics scheme that was first developed by Tzivion et al. (1987, 1989) and Feingold et al. (1988) with later applications and development documented in Stevens et al. (1996); Reisin et al. (1998); Yin et al. (2000a, b) and Rotach and Zardi (2007).

20

TAU differs from other bin microphysical codes because it solves for two moments of the drop size distribution in each of the bins rather than solving the equations for the explicit size distribution at each mass/size point, which allows for a more accurate transfer of mass between bins and alleviates anomalous drop growth.

In this version of the TAU microphysics<sup>3</sup>, the cloud drop size distribution is divided into 34 mass-doubling bins with radii ranging between 1.56 μm and 3200 μm. The method of moments (Tzivion et al., 1987) is used to compute mass and number concentrations in each size bin resulting from diffusional growth (Tzivion et al., 1989), collision-coalescence and collisional

<sup>&</sup>lt;sup>1</sup>http://weather.uwyo.edu/upperair/sounding.html

<sup>&</sup>lt;sup>2</sup>The acronym TAU refers to the Tel Aviv University, where it was primarily developed

<sup>&</sup>lt;sup>3</sup>Version available at https://www.esrl.noaa.gov/csd/staff/graham.feingold/code/ (Accessed on: 04/11/2017)

breakup (Tzivion et al., 1987; Feingold et al., 1988). Sedimentation is performed with a first-order upwind scheme. Aerosols are represented by a single prognostic variable that is assumed to have a log-normal distribution.

In addition, two bulk microphysics parameterizations were used to evaluate and analyze its performance: the schemes described by Thompson et al. (2008) and Morrison et al. (2009).

- The parameterization of Thompson et al. (2008) represents five hydrometeor species, namely, cloud water, rain, cloud ice, snow and graupel, which (except snow) are assumed to follow a gamma distribution. Snow is considered to be distributed according to an exponential function. It is a single-moment scheme, with the exception of the double-moment cloud ice variable. This scheme is based on Thompson et al. (2004), and it includes several improvements to various physical assumptions in an attempt to equate a full double-moment (or higher order) scheme. One of them is the introduction of an expression for the
- 10 gamma distribution shape parameter for cloud water droplets ( $\mu$ ) based on observations, which is specifically addressed in this work (Eq. 3).

$$\mu = \frac{1000}{N_d} + 2 \tag{3}$$

In Eq. 3,  $N_d$  represents the droplet concentration (cm<sup>-3</sup>), which has a fixed value (as in every single-moment parametrization), and should be defined by the users according to the mean conditions of the simulated case. In this scheme, an upper bound of

# 15 15 on the value of $\mu$ was defined.

20

The scheme of Morrison et al. (2009) predicts the mass mixing ratios and number concentrations (i.e., a double-moment scheme) of the same five hydrometeor species: cloud droplets, cloud ice, snow, rain, and graupel. The precipitation species, as well as cloud ice, follow an exponential size distribution. Meanwhile, cloud droplets are represented by a gamma distribution, where the  $\mu$  parameter is a function of the predicted droplet number concentration, following the observations of Martin et al. (1994) (Eq. 4, with minimum and maximum values of 2 and 10, respectively).

$$\mu = (5.714 \times 10^{-4} N_d + 0.2714)^{-2} - 1.$$
(4)

Both schemes use the following expressions to calculate  $\Lambda$  and  $N_0$ :

$$\Lambda = \left(\frac{\frac{\pi}{6}\rho_w N_d \Gamma(\mu+4)}{r_c \Gamma(\mu+1)}\right)^{\frac{1}{3}}$$
(5)

$$N_0 = \frac{N_d \Lambda^{\mu+1}}{\Gamma(\mu+1)} \tag{6}$$

where  $\rho_w$  represents the liquid water density (g m<sup>-3</sup>) and  $r_c$  is the cloud liquid water content (g m<sup>-3</sup>).

There is a difference in the way these two bulk parameterization schemes were used here: while the scheme of Thompson et al. (2008) was directly integrated into the KiD, to compare its results with those generated by the TAU, the scheme of Morrison et al. (2009) was incorporated just to consider a different approach to estimate gamma parameters. For this purpose,

30 we calculated the evolution of Gamma parameters according to the expressions used by the scheme in question but based on

the values of the DSD moments predicted by the TAU. The latter works as an error-corrected analysis, which allows focusing on the uncertainties introduced by the procedure for calculating the Gamma parameters. This method was also later applied to the approach of Thompson et al. (2008) for comparison.

# 2.2 Phase spaces

- If we consider a system consisting of a population of drops that follows a Gamma size distribution, then it is possible to track its evolution in the phase space determined by the three Gamma parameters ( $N_0$ ,  $\mu$  and  $\Lambda$ ). As each microphysics process produces different types of modifications in the shape of a DSD, the displacements in this "Gamma phase space", corresponding to the evolution of the DSD during the cloud life cycle, can be associated with specific combinations of those process intensities (Cecchini et al., 2017b).
- The Gamma phase space projection of the AC09 flight (RA1 in Cecchini et al. (2017b)) was taken as a reference to evaluate the performance of microphysics parameterizations. Cecchini et al. (2017b) obtained this projection by fitting a Gamma function to the DSD data measured by an airborne cloud droplet probe (Lance et al., 2010; Molleker et al., 2014) at the tops of growing convective clouds developed over the Amazon basin during the local dry-to-wet season transition. For more details on the AC09 flight, see Wendisch et al. (2016).
- The Thompson et al. (2008) and Morrison et al. (2009) bulk schemes determine the Gamma parameters in every time step; thus, its simulations can be directly projected into the Gamma phase space. However, for taking the simulations of the TAU scheme as a reference with respect to those schemes and compare it with the results of Cecchini et al. (2017b), a Gamma function must be fitted from its explicit size distribution. For coherence with the results of Cecchini et al. (2017b), we conserved the zeroth, second and third moments of the bin DSDs to obtain the three Gamma parameters (Equations 7, 8 and
- 9). We restricted our analysis to drops with diameters smaller than 50  $\mu$ m to avoid rain drops.

$$\mu = \frac{6G - 3 + \sqrt{1 + 8G}}{2(1 - G)} \tag{7}$$

$$\Lambda = \frac{(\mu+3)M_2}{M_3} \tag{8}$$

$$N_0 = \frac{\Lambda^{\mu+1} M_0}{\Gamma(\mu+1)}$$
 (9)

In equations 7, 8 and 9,  $M_p$  is the pth moment of the DSD, and G is the ratio:

$$G = \frac{M_2^3}{M_3^2 M_0} \tag{10}$$

The analysis of the results was performed in two phase spaces to provide a more comprehensive interpretation of the simulations: (a) the Gamma phase space and (b) the "bulk phase space". The latter is defined by two bulk properties of the DSDs: N<sub>d</sub>
(cm<sup>-3</sup>), which coincides with the zeroth moment of the DSD, and D<sub>eff</sub> (μm), which is the ratio between the third and second moments.

### 3 Results

### 3.1 Observation vs simulation

The Gamma phase spaces illustrated in Fig. 2 show the DSD evolution in the warm cloud that was simulated by the bin microphysics parameterization and the DSD evolution computed by Cecchini et al. (2017b) using measured DSDs. We tracked

- the evolution of the DSD at the top of the cloud for coherence with the AC09 sampling strategy. As already described, the simulation uses airplane and radiosonde data to reproduce nearly the same atmospheric state of those measurements. The large markers in Fig. 2b represent the averages for 200 m vertical intervals in the observation. Analogously, for time steps where the simulated cloud maintained the same maximum height, a mean cloud-top DSD was calculated. The growth of the modeled cloud is limited to lower heights compared to the observations because it includes only the warm-phase processes.
- Nevertheless, the progressive broadening of the DSDs is evidenced by the increase in  $N_0$  and the decrease in  $\Lambda$  and  $\mu$  in both cases.

Based on a Lagrangian, adiabatic assumption for cloud tops, Cecchini et al. (2017b) suggested that such a behavior of the trajectory in the Gamma phase space should be associated with the prevalence of collisional growth. However, in our simulations, following the top of the cloud, we are actually dealing with the DSDs resulting from a mixture of in-cloud and

15 environmental parcels. Moreover, we have checked that the effect of the collection is less important for the current stage of the simulated cloud.

The differences in absolute values between the graphics from Fig. 2 are determined by many factors. First, when dealing with the modeled cloud, the boundaries can be quantitatively defined; thus, there is more control over the path that follows the top of the cloud, as well as the position of the cloud base. Consequently, the initial portion of the graphic that represents the simulation

- includes information about the very beginning of the cloud, when the first droplets are activated and occupy only one or two bins of the DSD, while in the graphic that corresponds to the observation, the first DSDs plotted (lower heights above cloud base) correspond to a more developed stage of the cloud. This is why the simulated trajectory looks like an expanded version of the warm portion of the observed one. However, the qualitative similarity between the simulated and observed trajectories is quite remarkable, which ensures the bin microphysics simulation as a benchmark to study cloud processes and evaluate bulk
- parameterizations.

The description of the environmental conditions modulates the simulated DSD evolution and is also responsible for similarities and differences between the observed and simulated warm cloud evolution. For example, Fig. 3 shows that changes in the initial aerosol concentration can modify the position and shape of the simulated Gamma phase space trajectory by increasing the values of  $\Lambda$  and  $N_0$  as an expression of more numerous droplets and narrower DSDs.

For illustrating how the cloud-top trajectory in the Gamma phase space can be modified depending on the aerosol number concentration or other conditions, let us consider the evolution of the DSD inside one model grid point. Figure 4 shows the evolution of the DSD contained in the grid point located at a height of 1650 m above the surface for some arbitrary time steps of the simulation. Black circles represent the state of the system at the beginning of one model time step. The total displacement of the system for one time step of the model is determined by the results from advection and microphysics

processes. The microphysics algorithm is performed after the advection calculations; thus, to illustrate that sequence, the initial state for microphysics vectors is the final state for advection vectors in the figure. The main components of the microphysics effect, nucleation and condensation, are also represented. The collision-coalescence vector is not shown because, at the cloud stage being analyzed, its absolute value is several orders of magnitude smaller than that of the other vectors. In this figure, the vectors link the initial and final states corresponding to the action of each particular process, i.e., using the DSDs before and

5

after the execution of each algorithm.

At this stage of the simulation, the advection produces a sink effect by decreasing the droplet number concentration and the effective diameter of the initial DSD for each time step. Meanwhile, the two principal components of the microphysics, nucleation and condensation, act almost parallel to the  $N_d$  and  $D_{eff}$  axis in the bulk phase space, respectively. The condensational

- growth is high enough to increase the effective diameter of the DSD with respect to the initial state of the time step. However, 10 the nucleation is too weak to balance the diminution of the concentration caused by the advection. Consequently, the DSD evolves toward higher effective diameters and lower droplet number concentration. In the Gamma phase space, it translates mainly as a decrease in  $\Lambda$  with slight displacements toward higher values of  $N_0$  and smaller values of  $\mu$ .
- According to Cecchini et al. (2017b), the physical processes that modify the DSD can be referred as "pseudo-forces", in the 15 sense that they determine the direction and speed of the displacements forming the trajectories in these spaces. These pseudoforces are defined by properties such as the initial DSD, CCN characteristics, updraft speed, and supersaturation. For example, it is well known that higher aerosol concentrations enhance the nucleation and reduce the efficiency of condensational growth to increase the effective diameter of the DSD due to the water vapor competition.

Because each point of the graphs in Fig. 3 consists of the average for time steps where the cloud maintained the same maximum height, each cloud-top trajectory contains the average information from several segments, such as the one presented 20 in Fig. 4. If the correlation of forces for each of these segments is modified, then its path will change, and the average point will also be affected, thereby changing the overall shape and position of the cloud-top trajectory.

Figure 5a shows a bulk phase space containing time-averaged values of each cloud-top height for the same simulation. Three regimes can be identified: one showing an increase in  $N_d$  and  $D_{eff}$  with height in the lowest levels (label "1" in the figure); another in middle levels showing a strong increase in  $D_{eff}$  with constant  $N_d$  (label "2"); and the last one in upper levels where 25 only  $N_d$  varies significantly during cloud-top ascension (label "3"). This pattern is associated with changes in the tilting of the Gamma phase space trajectory in Fig. 2a and is related to variations in the relative intensity of advection and microphysics process rates, mainly nucleation and condensation.

30

Figures 5b-d illustrate the average effects of advection and microphysics processes at each of the mentioned regimes. In these figures, the vectors do not link two consecutive points because of two reasons: it is not a Lagrangian system, and we are dealing with average values for several time steps. However, such a vector approach is useful for analyzing the evolution of the force budget through the cloud-top history. In the lower altitudes (Fig. 5b), the occurrence of the maximum supersaturation favors droplet nucleation and condensational growth, which is more efficient at small droplet sizes. In the second regime, the nucleation vanishes; then, the DSDs are modified by condensational growth and advection from lower levels (Fig. 5c). Finally,

in the upper levels, the maximum vertical velocity enhances the contribution of the advection, while the condensational growth becomes negligible (Fig. 5d).

#### 3.2 Influence of the parameterization approach

In the previous section, we discussed how the evolution of the DSD, simulated by a bin microphysics scheme, can be modi-5 fied by different atmospheric conditions, and it was exemplified through the sensitivity to the aerosol number concentration. However, in bulk schemes, there is an additional source of uncertainties due to the particularity of using a pre-defined DSD function, whose flexibility depends on the number of moments being predicted. The correct description of the DSD is important because it controls several physical processes, such as droplet growth, evaporation and sedimentation. Therefore, different gamma parameters cause different bulk properties of the clouds, with consequences in the precipitation temporal and spatial

10 distributions.

> A common method in bulk parameterizations that uses gamma distributions for cloud droplets consists of fixing the  $\mu$ parameter. Because  $\Lambda$  and  $N_0$  depend on  $\mu$ , they also become limited. Figure 6 illustrates the simulation obtained from a bin microphysics parameterization, as described above, compared to the one from a bulk single-moment parameterization (Thompson et al., 2008) (hereafter "thompson08"). The Gamma phase space trajectories for both simulations are very different,

15 much more than the trajectories obtained from simulations with changes in the physical parameters of the bin scheme  $(N_a)$ , as was shown in the previous section. The thompson 08 parameterization defines  $\mu$  as a fixed parameter that is inversely proportional to the cloud droplet concentration, with a value between 2 and 15 (Eq. 3). However, that relation does not provide the DSD evolution during the cloud life cycle, as described by observations or simulated with the bin scheme (Sect 3.1). It occupies a completely different portion of the Gamma phase space, and its evolution direction is somehow opposite to that

from the bin scheme. This result occurs because, keeping  $\mu$  constant, the initially narrow DSD has to be represented by higher 20 values of  $N_0$ .

To avoid performing a comparison that involves accumulated errors, thus inducing larger differences in the DSD evolution, we also consider a hypothetical situation where, at every moment, the bulk scheme has the same values as the mixing ratio (and the concentration, when using a two-moment scheme) predicted by the bin scheme. This is also illustrated in Fig. 6 for

the thompson08 and morrison09 (Morrison et al., 2009) parameterizations ("bin-based" label in the legend). This figure shows 25 that, even if they were based on the correct values of the DSD moment(s), its DSD representation will be incorrect due to inefficiencies in the definition of the gamma parameters' dependence on those moments.

As explained in Sect. 2.2, for fitting a gamma distribution to the DSDs of the bin scheme, we use the  $0^{th}$ ,  $2^{nd}$  and  $3^{rd}$ moments. Thus, the value of  $\mu$  here should be a function of these variables, as defined by Eq. 7. However, Fig. 7a illustrates

that  $\mu$  is mostly determined by the magnitude of the  $3^{rd}$  moment. Then, we can approximate  $\mu$  at the cloud top as being 30 inversely proportional to the mixing ratio of cloud droplets  $(q_c)$ , which is conveniently the variable predicted by one-moment bulk schemes (Eq. 11). Note that Eq. 11 was defined by adding a  $q_c$ -dependent term to the expression for  $\mu$  originally employed

(11)

by the thompson08 scheme (Eq. 3).

$$\mu(q_c) = \frac{1}{q_c} + \frac{1000}{N_d} + 2$$

Defining  $\mu$  according to Eq. 11, without any modification to the way in which  $\Lambda$  and  $N_0$  are calculated, it is possible to reproduce the main characteristics of the bin simulation path in the Gamma phase space. This is illustrated in Fig. 7b, where

we have used Eqs. 5, 6 and 11 with the moments produced by the bin simulation to generate a bin-like path in the Gamma

phase space ("MOD" label in the figure).

The  $\mu$ -modified path in Fig. 7b, similar to the ones corresponding to the original approaches used in the thompson08 and morrison09 schemes in Fig. 6, was obtained from the moments predicted by the bin to avoid other types of errors that could exist on those parameterizations. To analyze the direct effect of the proposed modification in one-moment bulk parameterizations,

- we used the thompson08 scheme.  $N_d$  in Eq. 11 was defined as 700 cm<sup>-3</sup> for the thompson08 tests, as in Fig. 7a, but it must be variable if implemented in a two-moment scheme. Although the bin scheme can deal with extremely narrow DSDs, characterized by high values of  $\mu$  that appear at the beginning of cloud development, allowing such a variation in  $\mu$  does not perform well in the thompson08 scheme. Initially, small amounts of  $q_c$  would generate relatively high values of  $\mu$ ; the evolution of the DSD would then remain very limited, and no clouds would develop. The determination of the specific feature(s) of this
- scheme that could be responsible for such behavior is beyond the scope of this paper. For now, taking into account that the thompson08 scheme considers a variation of  $\mu$  between 2 for continental and 15 for maritime, according to the general dispersion characteristics from Martin et al. (1994) and the results of Cecchini et al. (2017b), we defined a threshold of 20 as an upper bound on  $\mu$  for the tests implemented here.

The effects of that modification on some bulk variables at the simulated cloud top are illustrated in Fig. 8. The droplet effective diameter and the rain-drop mixing ratio corresponding to the TAU simulation were inferred from the moments of the DSD that it predicts explicitly. In the case of the bulk scheme (original and modified), the droplet effective diameter was obtained from its gamma parameters, and the values of the rain-drop mixing ratio are the ones predicted explicitly by the scheme.

The new approach improves the bulk simulation through a reduction in the droplet effective diameter (Fig. 8a). This modification of the DSD has a positive effect on the temporal distribution of the rain-drop mixing ratio (Fig. 8b) by determining its rates of conversion from cloud droplets (autoconversion and collection). The cloud water mixing ratio remains unaltered because, at this stage, the amount that is being converted to rain is too small to cause an important sink effect and because, in this parameterization, the rates of cloud water production are not affected by the DSD shape.

Toward the interior of the cloud, the cloud-to-rain conversion rates should be larger. Then, a proportional decrease in droplet 30 growth rates would cause an increase in the cloud water content with respect to the original scheme, and an adjustment of 30 the rate of condensation would be necessary. Nevertheless, the expression for  $\mu$  that we are proposing mainly modifies the 30 beginning of the cloud development at each level (note that as  $q_c$  increases,  $\mu$  tends to a fixed value, determined by the last two 30 terms in Eq. 11). Improving the representation of the DSD at the cloud top would strongly impact the evolution of the cloud 30 given that it introduces a correction in the start point for each layer during cloud growth. Such a correction in the initial DSD

modulates the rates of microphysical process onward, determining the structure of the cloud. If the simulation continues, both warm phase processes and the ice processes would be affected, which depend on the DSD when dealing with phase transitions and mechanical interactions between ice and liquid water.

# 4 Concluding remarks

- 5 This paper documents the first attempt to take advantage of the potential of the Gamma phase space to evaluate and improve the performance of cloud microphysics parameterizations. This phase space allowed representing the simulated DSD evolution in a more comprehensive way and was useful for analyzing the associations between the displacements on it (representative of changes in the DSD with time and height) and the rates of microphysical processes.
- To validate the skills of a bin microphysics parameterization, the cloud-top trajectory in the Gamma phase space corresponding to in situ measurements was taken as a reference. This scheme was able to reproduce the main features of the observed DSD evolution, representing the progressive broadening of the DSDs as an increase in  $N_0$  and a decrease in  $\Lambda$  and  $\mu$  in the Gamma phase space. The simulation added information about earlier stages of cloud development thanks to the possibility of defining an objective criterion for the cloud initiation time and the position of its boundaries. These results allowed us to consider the bin microphysics parameterization as a valuable benchmark, useful for analyzing the dependence of the system responses
- on several parameters that characterize the environmental conditions and for evaluating the suitability of bulk microphysics approaches.

The agreement between the observed and simulated warm cloud evolution is determined by the description of the environmental conditions. We illustrated that the projections of the cloud top into the Gamma phase space are very sensitive to the aerosol concentration because of its influence over the correlation of pseudo-forces defining the DSD evolution on each layer. In

- this case, two general pseudo-forces determined the evolution of the DSDs for every grid point in the simulation: advection and microphysics, the latter being composed of nucleation and condensational growth. Differences in these pseudo-force budgets are responsible for variations in the evolution of cloud-top DSDs. Our results reinforce the idea that knowing the characteristics of the microphysics processes through an achievable domain in the Gamma phase space would provide a way to predict the evolution of a system from one initial state as a new approach in the development of microphysics parameterizations (Cecchini)
- et al., 2017b)

The Gamma phase space representation of the cloud top is highly affected by the approach chosen for the microphysics parameterization. Whereas the bin scheme approximately resembles the observations, bulk approaches generate completely different signatures, mainly in terms of its evolution direction. In an attempt to correct those deficiencies, we proposed an adjustment to bulk parameterizations based on the calculation of the  $\mu$  parameter according to the mixing ratio of cloud

droplets. The new approach provides a bin-like path in the Gamma phase space that corrects the cloud-top representation in bulk schemes. When this modification is introduced in the scheme of Thompson et al. (2008), the droplet effective diameter is reduced, with favorable consequences in the amount of precipitation.