# Peer review of "Cloud-top microphysics evolution in the Gamma phase space from a modeling perspective"

_Atmospheric Chemistry and Physics, 2018_

## Referee Comment (RC1) · Anonymous Referee #1 · 24 Mar 2018

This manuscript presents a numerical exercise that adjusted the gamma size distribution parameters in a bulk microphysics scheme based on the fitted gamma parameter values simulated by a 1D kinematic framework using the TAU bin microphysics scheme initiated by a sounding from the ACRIDICON-CHUVA (what is the full name it) campaign. The authors claimed that the so-called "gamma phase space" is useful for evaluating and improving microphysics schemes.

I found many aspects of the manuscript such as assumptions, concepts, logics, interpretations and presentations are questionable. I do not recommend this manuscript to be published in ACP. The main concerns and some technical comments are listed in

the following:

Major points:

1. The concept of using "pseudo forces" representing microphysical processes and cloud dynamics to explain the trajectory in "gamma phase space" (Cecchini et al. 2017) makes zero sense for the observational data. The cloud samples measured by the instruments at different times are combinations of hydrometeors experienced so many different microphysical and dynamical pathways. The derived gamma parameters based on these measurements are in no way determined by the "pseudo forces" that are only meaningful in a Lagrangian sense. It is ok to show the derived parameter values in the phase space. But it is not appropriate to interpret the relationships among them using the "pseudo force" concept. 2. As shown in McFarquhar et al. (2015), just one part of the observational uncertainties (counting uncertainty) lead to big ranges of gamma parameter values that describe the same equally realizable particle size distribution (PSD). The parameter uncertainty ranges can be comparable or greater than the differences between those derived by different measurement points, especially when PSD deviates from gamma distribution. 3. The authors may argue that McFarquhar et al. (2015) focused on mixed-phase while Cecchini et al. (2017) focused on liquid phase. However, the small range of the cloud droplet size (< 50 micron) should apply the incomplete gamma distribution fit rather than the complete gamma fit. Using the complete gamma fit results in higher uncertainties in the phase space. 4. The ranges of the fitted mu and lambda parameters based on TAU simulation span at least an order of magnitude wider in the phase space compared to the observed counterparts (Fig. 2). These high simulated values are unphysical and never observed. How can such unphysical representations of clouds serve as a base to improve the bulk microphyscs? 5. Bin microphysics schemes are conceptually more realistic but not practically more realistic. The bin microphysics intercomparison study by Xue et al. (2017) demonstrates that the uncertainties associated with bin microphysics schemes are similar if not greater than those in bulk schemes. A new study by Morrison et al. (2018) (JAS

under review now) shows that the combined advection in space (Eulerian framework) and in bin dimension (either mass or diameter) of the cloud droplet condensation process inevitably broadens the simulated PSD while the liquid water content and mean droplet diameter are accurately predicted. The model setup is similar to what is used in this work. The derived gamma parameters based on the TAU simulation did not correspond to the actual physics that lead to the observed values. 6. How is the cloud top defined in this work? All discussions and analysis are around "cloud-top" but no clear definition is stated. A profile plot of the initial relative humidity is helpful. The time evolution of the simulated cloud water and rain water profiles should also be provided (profiles of qc, nc, qr and nr in every 5 minutes would work). 7. Without knowing how the data were calculated in Fig. 5, I am still surprised to see the author claim that the advection increases the number and size of the cloud droplets in the cloud top (Fig. 5c and d). Where are the sources of these large droplets? 8. The mu-qc relationship (Fig. 7a) was found in the TAU data at cloud top. Was Eq. 11 applied to the Thompson scheme everywhere or just at cloud top in the simulation that generates Fig. 8? The mu-qc relationship outside of cloud top can be completely different. The observed mu-qc relationship can be very different than the bin results. The way to "improve" the bulk scheme does not necessarily need the gamma phase space concept.

Technical points:

1. Please add projections on the 3D plots. 2. More plots on the simulated cloud properties will be helpful.

References:

McFarquhar, G.M., Hsieh, T.L., Freer, M., Mascio, J. and Jewett, B.F., 2015. The characterization of ice hydrometeor gamma size distributions as volumes in N $0-\lambda-\mu$ phase space: Implications for microphysical process modeling. Journal of the Atmospheric Sciences, 72(2), pp.892-909.

Morrison, H., M. Witte, G. H. Bryan, J. Y. Harrington, and Z. J. Lebo, 2018: Spurious

broadening of modeled cloud droplet spectra using bin microphysics in an Eulerian spatial domain. JAS, in review.

Xue, L., Fan, J., Lebo, Z.J., Wu, W., Morrison, H., Grabowski, W.W., Chu, X., Geresdi, I., North, K., Stenz, R. and Gao, Y., 2017. Idealized Simulations of a Squall Line from the MC3E Field Campaign Applying Three Bin Microphysics Schemes: Dynamic and Thermodynamic Structure. Monthly Weather Review, 145(12), pp.4789-4812.
* * *

---

## Author Comment (AC1) · 3 Apr 2018

**Authors response to Anonymous Referee #1**

**(Comment)** This manuscript presents a numerical exercise that adjusted the gamma size distribution parameters in a bulk microphysics scheme based on the fitted gamma parameter values simulated by a 1D kinematic framework using the TAU bin microphysics scheme initiated by a sounding from the ACRIDICON-CHUVA (what is the full name it) campaign. The authors claimed that the so-called "gamma phase space" is

useful for evaluating and improving microphysics schemes. I found many aspects of the manuscript such as assumptions, concepts, logics, interpretations and presentations are questionable. I do not recommend this manuscript to be published in ACP. The main concerns and some technical comments are listed in the following:

**(Answer)** We would like to thank Anonymous Referee #1 for taking the time to analyze our work and pinpoint potential improvements. Here we provide a detailed response to the issues raised by Anonymous Referee #1. While we agree that the manuscript can be significantly improved based on the comments, we believe there is merit in publishing it given its novelty factor and the potential future research topics it enables.

**Major points:**

1. **(Comment)** The concept of using "pseudo forces" representing microphysical processes and cloud dynamics to explain the trajectory in "gamma phase space" Cecchini et al. (2017) makes zero sense for the observational data. The cloud samples measured by the instruments at different times are combinations of hydrometeors experienced so many different microphysical and dynamical pathways. The derived gamma parameters based on these measurements are in no way determined by the "pseudo forces" that are only meaningful in a Lagrangian sense. It is ok to show the derived parameter values in the phase space. But it is not appropriate to interpret the relationships among them using the "pseudo-force" concept.

1. **(Answer)** While the work of Cecchini et al. (2017) has its shortcomings, as any aircraft-based interpretation of DSD measurements, we definitely disagree with this first statement by Anonymous Referee #1.

Firstly, Amazonian clouds present a fairly uniform daily cycle: when you analyze animated satellite images for the region throughout several days, the "pulsing" aspect of convection is readily observable. Except on squall lines occasions (that can originate as far as the Atlantic Coast to the East), convection is primarily driven by differential heating of the surface (favored in regions with slight elevation – Machado et a. (2017))

[Figure]

and the cumulus field evolves from there. Therefore, when you plan an aircraft-based experimental campaign in the Amazon, you can choose to start flying by the time clouds are growing – note that ACRIDICON-CHUVA flights started in the late morning or early afternoon because of that. In this way it is possible to capture growing convective elements, especially when you primarily probe cloud tops – as done in ACRIDICON-CHUVA – and when you focus on updrafts – as done in Cecchini et al. (2017). The assumption that a snapshot of cloud-tops at various stages of the convection development equals the evolution of the top of an individual cloud that grows vertically was previously employed by Rosenfeld et al. (2008) to retrieve T-$r_e$ relationships that infer the vigor of severe convective storms.

What Cecchini et al. (2017) did was to use the altitude of DSD measurements as proxy for the time evolution of the clouds, which is justified given the specific configuration of the flight strategies. This proxy was used to generate the hypothesis that the DSD evolution can be understood as pseudo-forces in the Gamma phase space. This is as far as the Cecchini et al. (2017) could go, and no quantifications were provided because: 1) while the flight strategies serve as proxies for cloud evolution (in a semi-Lagrangean way) and certainly provide interesting insights for cloud DSD evolution, the uncertainties of the methodology itself impedes such quantifications; 2) the authors used a relatively simple conceptual model to study the trajectories – namely the condensation + collision-coalescence balance –, as a first step into the Gamma phase space, but other processes such as mixing should be considered for proper quantifications. Additionally, contemporary studies such as Yang et al. (2017) show that other processes may also contribute to DSD broadening aside from collision-coalesce as discussed in Cecchini et al. (2017). Our study aims to tackle exactly the shortcomings of Cecchini et al. (2017) by using a 1D model where there is more control over the conditions creating the DSDs being analyzed. In this way, the pseudo-forces approach can be directly tested.

2. **(Comment)** As shown in McFarquhar et al. (2015), just one part of the observational

uncertainties (counting uncertainty) lead to big ranges of gamma parameter values that describe the same equally realizable particle size distribution (PSD). The parameter uncertainty ranges can be comparable or greater than the differences between those derived by different measurement points, especially when PSD deviates from gamma distribution.

2. **(Answer)** As estimated by Cecchini et al. (2017), specifically in their Fig. 10, the observed DSDs used to generate the pseudo-forces hypothesis most likely evolve beyond the ellipsoids in the Gamma phase space proposed by McFarquhar et al. (2015) when we consider an instrument uncertainty of 10%. So, overlapping the corresponding ellipsoids through the entire path would allow to obtain the same conclusions about the DSD evolution. Also see the authors response to major point 6) of the Anonymous Referee #4 in the discussions of Cecchini et al. (2017):

https://www.atmos-chem-phys-discuss.net/acp-2017-185/acp-2017-185-AC4-supplement.pdf

In our case, a measure of the uncertainties contained in each point represented by big markers in Fig. 2a of the manuscript can come from the dispersion of the small markers that correspond to individual DSDs at each cloud-top height. It can be observed, analogously to the analysis of Cecchini et al. (2017), that the entire phase space trajectory evolves beyond the spread of the points at each level (symbolized by the color scale).

That being said, we believe there is a fundamental difference between our approach and the one proposed by McFarquhar et al. (2015) – both of them are valid and highlight different aspects of cloud microphysics.

The McFarquhar et al. (2015) approach originates from observational uncertainties and can supposedly help improve models with Monte Carlo assumptions in the future. While we agree that this is an adequate solution from a statistical/Monte Carlo standpoint, we disagree that it prevents a physically-based analysis such as the one

proposed here. The present work is not particularly interested in random movements in the Gamma phase space that originate from random DSD variabilities around a specific point. Rather, we want to understand what the physics behind the movements in the space is, which is equivalent to the knowledge of how one point moves from position A to position B. Additionally, when working with a deterministic model, there is no direct benefit on generating random movements to conform to observational characteristics. Instead, we focus on idealized trajectories in the phase space and how they can be related to microphysical processes.

To illustrate the complementary nature of both approaches, let us consider the following specific case. Figure 1 below roughly represents Fig. 5b from McFarquhar et al. (2015), where we substituted the ellipsoid projection with three straight lines (black continuous and dashed lines). The colors represent different effective diameter ($D_{eff}$, calculated as $\frac{\mu+3}{\Lambda}$) values in $\mu$m, where the region inside the black lines is highlighted. The solid line represents the major axis of the ellipsoid projection (note we limited the y axis to $3 \times 10^4$ m$^{-1}$ for clarity), while the dashed lines delimit upper and bottom boundaries.

A statistical view of Fig. 1 might highlight the spread in $D_{eff}$ values inside the ellipsoid, which can be associated to the underlying uncertainties of the application. However, a complementary and physically-based analysis might propose to understand what straight lines even mean in this space. It can be shown that every straight line conserves the averaged diameter $D_j$ of the form:

$$D_j = \frac{M_j}{M_{j-1}} = \frac{\mu + j}{\Lambda} \tag{1}$$

Where $M_j$ is the j'th moment of the DSD (when $j = 3$, $D_j = D_{eff}$). Eq. 1 basically states that the ratio between consecutive moments is conserved in straight lines in the $\mu - \Lambda$ space. The ratio between moments further apart should then be represented

by higher order polynomials. If we substitute $\Lambda = a + b\mu$ into Eq. 1, differentiate with respect to $\mu$ and equate the result to zero, we can find the moment ratio being conserved in any straight line by:

$$j = \frac{a}{b} \tag{2}$$

Using Eq. 2, we calculate $j = 2.2, 3.4, 4.6$ for the bottom, mid and top lines in Fig. 1. From this we conclude that the spread of $D_{eff}$ values inside the colored area stems from differences in the $a$ and $b$ coefficients. Those coefficients can also be understood as the degree of deviation from the exponential case and Eq. 2 can be rewritten as

$$j = \frac{\Lambda_{exp}\mu}{\Lambda - \Lambda_{exp}} \tag{3}$$

Where we note that $b = \frac{\Lambda - \Lambda_{exp}}{\mu}$ and $a = \Lambda_{exp}$ and $\Lambda_{exp}$ is the $\Lambda$ of an exponential function ($\mu = 0$). Another consequence from Eqs. 2 and 3 is that it is possible to fix $j$ and find all the lines that conserve $D_j$, thus producing contour lines similar to Fig. 1 and ultimately a continuous surface.

All of the explanations above have the intention of highlighting the possibilities of treating the Gamma phase space as a physical entity as well as a statistical one. Additionally, it shows that there are preferential shapes on the phase space depending on the underlying physics of the problem being analyzed. The ellipsoid shape is justified in observational studies because of the proximity of the DSDs in the phase space. However, different points inside the ellipsoids may be related to different physical processes, or different "histories" that drove the points there. Further analysis that better explain the correlations between the Monte Carlo and deterministic approaches in the Gamma phase space are surely encouraged.

3. **(Comment)** The authors may argue that McFarquhar et al. (2015) focused on mixed-

phase while Cecchini et al. (2017) focused on liquid phase. However, the small range of the cloud droplet size ($< 50 \mu m$) should apply the incomplete gamma distribution fit rather than the complete gamma fit. Using the complete gamma fit results in higher uncertainties in the phase space.

3. **(Answer)** The complete/incomplete Gamma discussion was added to the Cecchini et al. (2017) paper as part of the revision process. See the discussion in the major point 4) here:

https://www.atmos-chem-phys-discuss.net/acp-2017-185/acp-2017-185-AC5-supplement.pdf

The overall conclusion is that the relation between different DSD moments is conserved between the incomplete and complete approaches, even though the values of the moments themselves are different. In other words, the complete- and incomplete-Gamma trajectories are most likely parallel and so are the pseudo-forces.

We mentioned the 50 $\mu$m restriction for droplets diameter to be coherent with the analysis of Cecchini et al. (2017), however, there is no significant quantity of rain drops in the simulated cloud-top. Figure 2 shows the DSDs for some cloud-top heights, averaged for time-steps where the model stayed in the same maximum height. It can be seen that, for diameters larger that 30 $\mu$m, there are less than one droplet per cm$^3$, actually a few droplets per m$^3$.

4. **(Comment)** The ranges of the fitted $\mu$ and $\Lambda$ parameters based on TAU simulation span at least an order of magnitude wider in the phase space compared to the observed counterparts (Fig. 2). These high simulated values are unphysical and never observed. How can such unphysical representations of clouds serve as a base to improve the bulk microphysics?

4. **(Answer)** The fact that high values of $\mu$ and $\Lambda$ (and low values of $N_0$) are not observed does not necessarily mean they are unphysical. It may only mean that our

current instrument setup does not detect them. Note that those values occur close to cloud base and they get closer to the observed values as the cloud grows. As commented in the manuscript (section 3.1, third paragraph), the observed droplets existed for much longer than the modeled ones, otherwise the cloud would not even be visible from the airplane. The definition of an objective threshold to define the cloud boundaries (see answer to question 6) allows us to sample cloud stages that may not be considered in the observations. Moreover, Cecchini et al. (2017) showed a 200m-averaged data, while we use a 50m grid-spacement, therefore representing a more detailed path in the gamma phase space.

The question now is: what physical processes produce such high $\mu$ and $\Lambda$ values? As mentioned before, this type of DSD appears mostly close to cloud base and the high shape and curvature parameters produce very narrow distributions. They are basically a result of freshly activated CCNs, where the droplets cover only a small number of bins. In other words, the cloud is just being formed and there was not enough time for growth mechanisms to occur.

The latter is illustrated in Fig. 3. It shows the actual DSD and the fitted gamma function corresponding to the first point in the path of Fig. 2a of the main manuscript.

The sampling strategy determines the localization of the curve in the gamma phase space. So it is possible to increase the similarity between the observed and simulated paths if we modify the cloud-top definition and the minimum altitude above cloud-base at which we start to track the cloud-top in the simulation. Consequently, moving the trajectory inward or outward the cloud avoids the extreme, questioned values that correspond to non-easily observed stages of cloud development (Fig. 4). We have to keep in mind that the model setup is highly idealized, therefore its prognostics are not meant to be quantitatively precise. Conversely, the intent of using such tools is to study specific processes that can help build a conceptual model rather than trying to faithfully reproduce nature. The important thing to note, independently of the sampling strategy in the simulation, is the qualitative similarity between the modeled and observed

trajectories, and the reasons for that should be explored.

As we mentioned in the sixth paragraph, section 3.2, we are aware that such high values of $\mu$ obtained at the beginning of the TAU cloud does not perform well in the bulk scheme, possibly because this kind of schemes include more observation-based features than bin schemes. For that reason, we restricted the expression for $\mu$ that was applied, citing the manuscript: "For now, taking into account that the thompson08 scheme considers a variation of $\mu$ between 2 for continental and 15 for maritime, according to the general dispersion characteristics from Martin et al. (1994) and the results of Cecchini et al. (2017), we defined a threshold of 20 as an upper bound on $\mu$ for the tests implemented here." We showed and discussed the entire evolution in the gamma space, but we did not use those extreme and somehow questionable values of $\mu$, $\Lambda$ and $N_0$ in the final expression for $\mu$.

5. **(Comment)** Bin microphysics schemes are conceptually more realistic but not practically more realistic. The bin microphysics intercomparison study by Xue et al. (2017) demonstrates that the uncertainties associated with bin microphysics schemes are similar if not greater than those in bulk schemes. A new study by Morrison et al. (2018) (JAS under review now) shows that the combined advection in space (Eulerian framework) and in bin dimension (either mass or diameter) of the cloud droplet condensation process inevitably broadens the simulated PSD while the liquid water content and mean droplet diameter are accurately predicted. The model setup is similar to what is used in this work. The derived gamma parameters based on the TAU simulation did not correspond to the actual physics that lead to the observed values.

5. **(Answer)** We understand that bin models, despite having a higher accuracy on the representation of physical processes, may not be able to fully represent clouds in nature. They still have their internal assumptions and the usually idealized input data prevents realistic representations. What the Xue et al. (2017) paper notes is that different bin schemes produce similar systems overall, that have significantly different internal structure. One of the main reasons for that was pointed out to be the representation of ice processes. Note that our present work deals exclusively with warm processes, where the processes are much less complex. Either way, we are not trying to precisely represent the observed cloud – otherwise we would not have chosen a 1D model. The idea behind the model choice is to have a tool that precisely calculates the baseline physical processes, while limiting complex and non-linear interactions between microphysics and dynamics. The aim is to establish the basic concepts on the interpretation of the Gamma phase space.

We cannot comment directly on the Morrison et al. (2018) paper, or attest to its methodology, because we have no access to it. However, we also noted the pattern of widening DSDs even with little or no collision-coalescence growth. Figures 4 and 5 (main manuscript) address this issue by segregating the pseudo-forces. Taking into account the processes resolved by the model, we noted two mechanisms that can widen the DSD during the early stages of the cloud where there is no collection growth yet. Both advection and condensational growth were at least partly responsible for the DSD widening in our case, determining a path in the gamma phase space that closely resembles the observed in Cecchini et al. (2017). We will surely look forward for the publication of the Morrison et al. (2018) paper to study additional features that can help explain our results – particularly the advection in bin dimension.

6. **(Comment)** How is the cloud top defined in this work? All discussions and analysis are around "cloud-top" but no clear definition is stated. A profile plot of the initial relative humidity is helpful. The time evolution of the simulated cloud water and rain water profiles should also be provided (profiles of $q_c$, $n_c$, $q_r$ and $n_r$ in every 5 minutes would work).

6. **(Answer)** Figure 5 shows $q_c$ (g/kg), $N_c$ (cm$^{-3}$), $D_{eff}$ ($\mu$m) and relative humidity (RH) (%) for the entire simulation. See that the upward advection causes a maximum of $N_c$ at cloud-top for all times. As droplets ascend and mix with new droplets, they grow by diffusion of vapor and, to a lesser extent, by collision-coalescence. As a consequence, $D_{eff}$ and $q_c$ are larger in upper levels at the last times of the simulation.

[Figure]

There is no rain class in the TAU scheme, instead, the bins are intended to span through all the liquid water spectrum. The above figures include the information from all the bins, so they theoretically include both cloud and rain droplets. Defining a size threshold, it would be possible to discriminate between rain and cloud water. However, as can be observed in Fig. 2, there is no significant quantity of rain drops in the cloud-top. So Fig. 5 represent basically cloud droplets.

The cloud-top was defined as the last model level, from surface to top, where the droplets concentration was larger than 100 per cm$^3$. As we commented in the manuscript (section 3.1, fourth paragraph), variations in environmental characteristics, exemplified by the aerosol number concentration (Fig.3 in the manuscript), changes the localization of the path in the gamma phase space. A similar effect can be obtained if we vary the droplets number concentration threshold that defines cloud-boundaries. The variations of the restricted path in the gamma phase space are shown in Fig. 4. Despite several aspects modulates the similarity of the simulated gamma path with the measured values of Cecchini et al. (2017), they keep the same trend to move from high $\Lambda$ and $\mu$, and small $N_0$, toward smaller $\Lambda$ and $\mu$, with higher $N_0$, which represents the evolution from incipient to more developed DSDs in both simulation and observations. Again, our intention was not to obtain accurately coincident paths, we wanted to emphasize the general trend that has to be guaranteed in order to be physically coherent.

The definition of cloud-top, as well as the time evolution of $q_c$, $N_c$ and $D_{eff}$ were added to the manuscript.

7. **(Comment)** Without knowing how the data were calculated in Fig. 5, I am still surprised to see the author claim that the advection increases the number and size of the cloud droplets in the cloud top (Fig. 5c and d). Where are the sources of these large droplets?

7. **(Answer)** The markers in Figure 5a (main manuscript) represents the same data

that big markers in Fig. 2a (main manuscript): time-averages of the cloud-top DSDs for the times-steps when the cloud-top stayed at the same level of the model vertical domain, but in this case for two bulk properties of the DSD, instead of the three parameters of the fitted gamma function. Analogously, we constructed Fig. 5b,c and d (main manuscript), limited to some points within each regime defined in Fig. 5a (main manuscript). The vectors were represented in the same way than in Fig. 4 (main manuscript): linking initial and final stages due to each microphysics or advection processes, but in the case of Fig. 5 (main manuscript), they represent averaged initial and final states for the times the cloud-top remained in the same level, i.e. they represent the time-averaged effect of each process for a constant cloud-top height. We added this explanation to the manuscript, for clarity.

We commented in section 3.1, sixth paragraph, regarding the information represented in Fig. 4 (main manuscript), that the advection produced a sink effect. That occurs because it refers to a point that is fixed at 1650 m height above surface, and therefore near to cloud-base. At cloud-base, the content that the advection mechanism takes away surpasses what it brings from the inferior layer. Conversely, at upper levels, such as the ones represented in Fig.5c and d (main manuscript), there is more content to bring from below the layer, and the advection produces a net source effect. To increase the effective diameter, there is no need for larger drops, a bigger quantity of the largest ones that already exist is sufficient. Also, note that the effective diameter is the ratio between two DSD moments of consecutive order (the second and third moments), the higher one being in the numerator. The higher the order of the moment, the more weight for larger droplets. So, if we increase the same amount of droplets for every bin, higher order moments will increase faster than smaller ones. Therefore, the effective diameter can increase even if every bin number concentration increases proportionally.

8. **(Comment)** The $\mu$-$q_c$ relationship (Fig. 7a) was found in the TAU data at cloud top. Was Eq. 11 applied to the Thompson scheme everywhere or just at cloud top in the simulation that generates Fig. 8? The $\mu$-$q_c$ relationship outside of cloud top can be

completely different. The observed $\mu$-$q_c$ relationship can be very different than the bin results. The way to "improve" the bulk scheme does not necessarily need the gamma phase space concept.

8. **(Answer)** Yes, the $\mu$-$q_c$ relationship outside of cloud-top can be completely different. Because the $\mu$-$q_c$ was found in the TAU data at cloud-top, we agree that, a priori, it should only be applicable at cloud-top. Note that Eq. 11 tends to Eq. 3 when $q_c$ tends to infinity, as we explain in the manuscript (section 3.2, fourth paragraph), because we just added a term that is inversely proportional do $q_c$. Therefore, toward the interior of the cloud, as $q_c$ increases, the introduced modification stops making a difference, not affecting the way $\mu$ was previously determined in thompson08 scheme (also explained in section 3.2, ninth paragraph).

We concur with Anonymous Referee #1 that the observed $\mu$-$q_c$ relationship can be very different than the one obtained from bin schemes. However, a common problem in modelling microphysics processes comes from the difficulty to obtain direct measurements of hydrometeors to improve theory and to perform direct comparisons, which brings us back to item 1 of this document. As a consequence, microphysics parameterizations has to be evaluated based on secondary quantities, such as precipitation estimated from remote sensors, etc. On the other hand, despite its shortcomings, bin schemes are considered more realistic because they use a reduced number of simplifications with respect to bulk approaches. As a consequence, they are usually considered as a reference to adjust bulk parameterizations. The proposed $\mu$-$q_c$ relationship satisfies the objective of inducing an already validated bin feature –the cloud-top trajectory in the gamma phase space– into a simpler scheme that was proven to misrepresent it. So we are not only reproducing a random characteristic of a bin scheme, we are trying to bring an observed particularity to a bulk scheme, using the bin parameterization as a tool.

The phase space concept is a tool, as many others usually addressed in science to better visualize and understand physical processes. Maybe we could arrive to the same

result without mentioning the gamma phase space, but it would increase the difficulty of otherwise simpler interpretations. What is actually needed is a correct description of gamma parameters, without it almost all microphysics calculations remains unrealistic and unphysical. We believe that characterizing the gamma phase space mathematically and physically is a worthwhile step in that sense.

**Technical points:**

1. **(Comment)** Please add projections on the 3D plots.

1. **(Answer)** Projections were added in all 3D plot, except for Fig. 4b, where projections would hamper the visualization.

2. **(Comment)** More plots on the simulated cloud properties will be helpful.

2. **(Answer)** Plots on the simulated $q_c$, $N_c$ and $D_{eff}$ were included in the manuscript.

[Figure]

**Fig. 1.** An adaptation of Fig. 5b in McFarquhar et al. (2015)

[Figure]

[Figure]

**Fig. 2.** Examples of the simulated cloud-top DSDs

**Fig. 3.** Droplet size distribution corresponding to the point that is closer to cloud-base in Fig. 2a of the main manuscript

[Figure]

**Fig. 4.** Comparison between the observed trajectory and the modeled ones, using different sampling strategies and avoiding the youngest stages of the cloud.

[Figure]

(a) $q_c$ (g/kg)          (b) $N_c$ (cm$^{-3}$)

[Figure]

(c) $D_{eff}$ ($\mu$m)          (d) RH (%)

**Fig. 5.** Evolution of cloud properties profiles for the TAU simulation

**Supplement:**

[revised manuscript text omitted]
 simulation with 800 aerosol particles per cm$^3$, i.e. the same data that big markers in Fig. 3a. Three regimes can be identified: one showing an increase in $N_d$ and $D_{eff}$ with height in the lowest levels (label "1" in the figure); another in middle levels showing a strong increase in $D_{eff}$ with constant $N_d$ (label "2"); and the last one in upper levels where only $N_d$ varies significantly during cloud-top ascension (label "3"). This pattern is associated with changes in the tilting of the Gamma phase space trajectory in Fig. 3a and is related to variations in the relative intensity of advection and microphysics process rates, mainly nucleation and condensation.

Figures 6b-d illustrate the average effects of advection and microphysics processes at each of the mentioned regimes. The vectors were constructed in the same way as in Fig. 5: linking initial and final stages due to each microphysics or advection processes, but in the case of Fig. 6, they represent averaged initial and final states for the times the cloud-top remained in the same level, i.e. they represent the time-averaged effect of each process for a constant cloud-top height. 
[revised manuscript text omitted]

*Acknowledgements.* This research was funded by the SOS CHUVA FAPESP Project 2015/14497-0. The contributions of Micael A. Cecchini and Lianet H. Pardo were funded by FAPESP grants 2017/04654-6 and 2016/24562-6, respectively.

[Figure]

**Figure 1.** Model configuration: (a) initial conditions and (b) prescribed field of vertical velocity

[Figure]

**Figure 2.** Evolution of $N_d$ (cm$^{-3}$), $D_{eff}$ ($\mu$m) and $q_c$ (g/kg) in the simulation. The black lines represent cloud-top.

[Figure]

(a)                                                    (b)

**Figure 3.** Gamma phase space representation of cloud-top DSDs for different cloud widths: (a) bin microphysics simulation and (b) observation (Fig. 6 of Cecchini et al. (2017b)). Small markers represent 1 Hz data, while larger ones are averages for every model level in the simulation and for 200 m vertical intervals in the observation. The color scale represents the height above the cloud base in meters. Projections on axis planes are represented by black continuous lines, in the simulation, and dashed lines, in the observation.

[Figure]

**Figure 4.** Illustration of the sensitivity of cloud-top DSDs to the initial aerosol number concentration in the Gamma phase space. The markers represent the average DSDs for each model level. Projections on axis planes are represented by continuous lines.

[Figure]

**Figure 5.** Forces governing the evolution of one grid-point DSD in the simulation: advection (Adv) and the results from all microphysics processes (Mphys), composed mainly of nucleation (Nuc) and condensation (Cond) in (a) the bulk phase space and (b) the Gamma phase space

[Figure]

**Figure 6.** Bulk phase space representation of cloud-top DSDs: (a) for all the stages of the simulated cloud development (the color scale represents the height above the cloud base in meters); (b), (c) and (d) for some points in low, middle and high cloud-top heights, respectively. The vectors represent the time averages of $\Delta N_d$ vs $\Delta D_{eff}$ at each level due to the advection (Adv) and the results from all microphysics processes (Mphys), mainly nucleation (Nuc) and condensation (Cond).

[Figure]

**Figure 7.** Phase space representation of the cloud-top layer when using TAU and thompson08 parameterizations, and the thompson08 and morrison09 approaches based on the moment(s) predicted by the TAU (bin-based). Projections on axis planes are represented by continuous lines.

[Figure]

**Figure 8.** (a) Relation between $\mu$ and $q_c$ as obtained from the bin simulation cloud-top DSDs, (b) Gamma phase space representation of the cloud-top DSDs using a bin parameterization (TAU) and a modified approach for application in bulk schemes (MOD) based on the moment(s) predicted by the bin parameterization. The color scale corresponds to the height in meters. Projections on axis planes are represented by continuous lines, black ones for TAU and red ones for MOD.

[Figure]

**Figure 9.** Comparison of the evolution of cloud-top properties in bin and bulk simulations before and after modifying $\mu$: (a) droplet effective diameter and (b) rain-drop mixing ratio

[revised manuscript text omitted]

35 simulation with 800 aerosol particles per cm$^3$ , i.e. the same data that big markers in Fig. 3a. Three regimes can be identified:

one showing an increase in $N_d$ and $D_{eff}$ with height in the lowest levels (label "1" in the figure); another in middle levels showing a strong increase in $D_{eff}$ with constant $N_d$ (label "2"); and the last one in upper levels where only $N_d$ varies significantly during cloud-top ascension (label "3"). This pattern is associated with changes in the tilting of the Gamma phase space trajectory in Fig. 23a and is related to variations in the relative intensity of advection and microphysics process rates, mainly nucleation and condensation.

Figures 56b-d illustrate the average effects of advection and microphysics processes at each of the mentioned regimes. The vectors were constructed in the same way as in Fig. 5: linking initial and final stages due to each microphysics or advection processes, but in the case of Fig. 6, they represent averaged initial and final states for the times the cloud-top remained in the same level, i.e. they represent the time-averaged effect of each process for a constant cloud-top height. 
[revised manuscript text omitted]

*Acknowledgements.*   This research was funded by the SOS CHUVA FAPESP Project 2015/14497-0. The contributions of Micael A. Cecchini and Lianet H. Pardo were funded by FAPESP grants 2017/04654-6 and 2016/24562-6, respectively.

[Figure]

**Figure 1.** Model configuration: (a) initial conditions and (b) prescribed field of vertical velocity

[Figure]

**Figure 2.** Evolution of $N_d$ (cm$^{-3}$), $D_{eff}$ ($\mu$m) and $q_c$ (g/kg) in the simulation. The black lines represent cloud-top.

[Figure]

(a)                                                                (b)

**Figure 3.** Gamma phase space representation of cloud-top DSDs for different cloud widths: (a) bin microphysics simulation and (b) observation (Fig. 6 of Cecchini et al. (2017b)). Small markers represent 1 Hz data, while larger ones are averages for every model level in the simulation and for 200 m vertical intervals in the observation. The color scale represents the height above the cloud base in meters. Projections on axis planes are represented by black continuous lines, in the simulation, and dashed lines, in the observation.

[Figure]

**Figure 4.** Illustration of the sensitivity of cloud-top DSDs to the initial aerosol number concentration in the Gamma phase space. The markers represent the average DSDs for each model level. Projections on axis planes are represented by continuous lines.

[Figure]

**Figure 5.** Forces governing the evolution of one grid-point DSD in the simulation: advection (Adv) and the results from all microphysics processes (Mphys), composed mainly of nucleation (Nuc) and condensation (Cond) in (a) the bulk phase space and (b) the Gamma phase space

[Figure]

**Figure 6.** Bulk phase space representation of cloud-top DSDs: (a) for all the stages of the simulated cloud development (the color scale represents the height above the cloud base in meters); (b), (c) and (d) for some points in low, middle and high cloud-top heights, respectively. The vectors represent the time averages of $\Delta N_d$ vs $\Delta D_{eff}$ at each level due to the advection (Adv) and the results from all microphysics processes (Mphys), mainly nucleation (Nuc) and condensation (Cond).

[Figure]

**Figure 7.** Phase space representation of the cloud-top layer when using TAU and thompson08 parameterizations, and the thompson08 and morrison09 approaches based on the moment(s) predicted by the TAU (bin-based). Projections on axis planes are represented by continuous lines.

[Figure]

**Figure 8.** (a) Relation between $\mu$ and $q_c$ as obtained from the bin simulation cloud-top DSDs, (b) Gamma phase space representation of the cloud-top DSDs using a bin parameterization (TAU) and a modified approach for application in bulk schemes (MOD) based on the moment(s) predicted by the bin parameterization. The color scale corresponds to the height in meters. Projections on axis planes are represented by continuous lines, black ones for TAU and red ones for MOD.

[Figure]

**Figure 9.** Comparison of the evolution of cloud-top properties in bin and bulk simulations before and after modifying $\mu$: (a) droplet effective diameter and (b) rain-drop mixing ratio

---

## Referee Comment (RC2) · Anonymous Referee #2 · 4 Apr 2018

This is a review of the manuscript "Cloud-top microphysics evolution in the Gamma phase space from a modeling perspective" by Pardo et al., submitted to ACPD. The authors are interpreting gamma drop size distributions (DSDs) in terms of the parametrs N0, lambda, mu, in addition to the moments of the DSD. Measured DSD properties at the cloud top of a convective plume are compared to model simulations, using both bin and bulk microphysics schemes. According to the authors, the proposed method of interpretation yields additional insight into model results to provide a better understanding of cloud microphysical processes, including aerosol cloud interactions. The outcome of the study is a modified parameterization of the cloud droplet shape parameter, as commonly used in two-moment bulk microphysical models.

Personally, after reading this manuscript (and parts of its precursor, Cecchini et al. 2017) I am still having a hard time to see the point of using the "gamma phase space" for interpretations of physical processes, or even just to compare differences between models and/or measurements. By recognizing the units of the parameters, it is obvious that the physical interpretation of these parameters is far less straightforward or intuitive than looking at the moments and the corresponding change rates - number, mass, and maybe surface or reflectivity if we want to add a third moment (thus constraining all three parameters N0, lambda, mu). In the end, the authors seem to set aside their previously introduced method, and use the moments or a combination of moments instead, stating it yields additional physical insight. Large parts of the results section provide only short descriptions of what the parameters look like in the plots, while the actual interpretation is done based on the moments. I cannot see which conclusion of the paper would not have been possible by looking at the moments only. Also the outcome of their new parameterization is a function of bulk number and mass (eq. 11). So why should I make an additional effort in future and explicitly interpret the parameters, and why should I call a process rate pseudo force?

A new parameterization of mu is proposed, and since the manuscript advertises the gamma phase space I was expecting something like mu being a function of other parameters - e.g., for rain drops it is common practice that mu=fct(lambda). This is not even mentioned, instead the authors continue to rely on moments of the DSD. While I am not saying it is a bad idea, I cannot see how the gamma phase space has contributed here. Unfortunately, there is no indication of whether the new parameterization would be applicable to any other situations.

Another example is how the manuscript addresses the effect of aerosols on cloud properties. Figure 3 shows the sensitivity of the parameters to aerosol concentrations. The main dependency seems to be represented by the magnitude of N0 (while it is really challenging to see even qualitative dependencies in the 3d plots). N0 is proportional to the bulk number (zeroth moment), but at the same time a complicated function of
mu and lambda (which already span the other two dimensions). So the only effect of substituting bulk number by N0 is that the analysis becomes more complicated or even meaningless - personally I cannot calculate (lambda(mu+1)/gamma(mu+1)) without using tools. On the other hand, the dependency of bulk number on aerosol concentration is well-established.

In the discussions about forces I do not understand why advection is considered one of them. When I imagine to be sitting within a parcel below cloud top that is being transported upward: Why would advection impose any changes on me, while I am moving along with the parcel and my direct environment as well? I am not resting at one level.

The main message I am taking away is that current parameterizations of the cloud droplet shape parameter, oftentimes a function of droplet number, are probably not in a final stage yet and there is room for improvements. This confirms what was recently described by Igel and van den Heever (2017, DOI: 10.1175/JAS-D-15-0382.1). At the same time, the possibility of using 3-moment bulk schemes to explicitly predict the shape parameter based on the microphysical processes is hardly mentioned in the manuscript.

As part of the discussion within ACPD, I also want to comment on the criticism of Reviewer 1. The paper of Xue et al. (2017) is cited in order to establish that in practice, bin microphysical models may not be useful because they yield a spread in the results that is comparable to bulk models. I am going to explain why I cannot agree with the reviewer's opinion who claims that we cannot trust the bin model used here: Xue et al. (2017) present a model intercomparison of three bin models which simulate a squall line in an idealized setup. They find considerable differences which are solely attributable to the microphysical processes and their representation. However, the very point of the paper is that even bin models – whose primary advantage is a free evolution of the particle size distribution – are still relying on and suffering from a number of assumptions related to ice microphysics such as particle densities, shapes,

**ACPD**
conversion thresholds, treatment of liquid fractions, etc. On the other hand, liquid-only microphysics are way less ambiguous, even though we can think of slightly different relations for fall velocities, coalescence efficiencies and other details. Therefore the heavy criticism seems inappropriate.

While I regret I could not extract the essence of the manuscript regarding the gamma phase space, I am going to provide a number of specific suggestions for improvements to the manuscript. Generally, I find a lot of vague sentences and I wished there were more information and more specific sentences in all parts of the manuscript.

Page 1 Lines 22,23: What are practical applications? "Generally employed" is very vague or even wrong. Also references might help.

Line 5: N has units of cm-3 only when integrated over a finite size interval. There seem to be inconsistencies with units also in other places, see below.

Lines 6-10: vague formulation, what are "enough" moments. The impact of mu on cloud water path and condensation rates are described, but which mu are we talking about – cloud, rain, ice? Three-moment schemes have been introduced more than 10 years ago and it would be appropriate to mention at this point.

Lines 13-18: Very vague: What is very useful about it, what are the specific advantages? What are the new opportunities?

Page 3 Line 8: what is the sounding date, also there is a lack of information about the AC09 flight. It is cited further below but still it would be nice to have a quick overview including date and the cloud we are looking at.

Page 4 Line 2: What does the prognostic variable represent? What are the initialized aerosol properties in the model?

Lines 5-10: The statements about ice properties seem to be irrelevant for this study. Since mu is a central topic here, what are the underlying observations/ cloud types/ reference other than Thompson?

**ACPD**
Line 18: It seems worth noting that the exponential is a gamma distribution with mu=0.

Line 26: The morrison scheme uses SI units. Also it does not use mass densitites, but mixing ratios.

Lines 29 and following: I do not understand: What is different in the approach of Morrison to estimate the parameters? Line 22 states that both schemes use the same expressions. I am also curious whether potential differences between the schemes in terms of units are considered correctly. Trying to understand: The Morrison scheme is not used to calculate any process rates, but only to diagnose DSD parameters? Are the expected differences due to the parameterization of mu or anything else? If so, why not simply replace the parameterization of mu within the Thompson scheme with the one used in the Morrison scheme? But in contrast to the Morrison scheme, the Thompson microphysics does calculate its own process rates? How can the comparison be fair when Morrison gets the moment input from TAU, but Thompson predicts the moments on its own? What are "the uncertainties introduced by the procedure..." – only the parameterization of mu?

Page 5 Line 8: Please explain "process intensities", or otherwise it would be helpful to stick to established wording.

Lines 10-14: What makes the phase space a projection? Is is something different from the 3d space that is used here?

Line 20: Please explain the restriction: Are there considerable amounts of drizzle/rain present which is just cut off from the DSD? Is the intention to avoid having a second mode in the DSD, or are there other reasons? 50 micron appears pretty small indeed. Even though most of the bulk number will be contained at sizes below this threshold, considerable fractions of mass can be contained in the tail larger than that. This may be also a reason for the big values of diagnosed shape parameters.

Lines 28 - 31: The "bulk phase space" is another example when I feel that new content
is created by new wording only. The authors interpret the moments of the DSD in order to get an idea about the physical processes, which has been done by the community for decades. Since the authors also see the need to do so, I am concluding that the "gamma phase space" as such is limited in being useful to interpret the physics.

Page 6: Line 9: There seems to be a hint that the real cloud contained ice, but the model does not? What does it mean that the cloud was limited to lower heights in the model?

Line 10: Mu is commonly referred to as the width of the DSD – isn't it a sufficient criterion for a broadening DSD to find a decrease in mu? How are N0 and lambda important in interpreting the broadening? Could we also think of opposite tendencies for N0/lambda and still call it broadening?

Figures: Is log() referring to the natural logarithm? In particular to interpret the numerical values of mu, log10() scales will be much more intuitive. At the same time I wonder if the values considerably larger than, say 20-30, are pointing to problems in the diagnosis of mu. The 3d plots provide hardly any usable information, even qualitative judgements are difficult. It would make sense to show the data under discussion in a 2d plane or some other kind of restructuring. **ACPD**

---

## Author Comment (AC2) · 26 Apr 2018

(**Comment**) This is a review of the manuscript "Cloud-top microphysics evolution in the Gamma phase space from a modeling perspective" by Pardo et al., submitted to ACPD. The authors are interpreting gamma drop size distributions (DSDs) in terms of the parameters  $N_0$ ,  $\Lambda$ ,  $\mu$ , in addition to the moments of the DSD. Measured DSD properties at the cloud top of a convective plume are compared to model simulations, using both bin and bulk microphysics schemes. According to the authors, the proposed method of interpretation yields additional insight into model results to provide a better understanding of cloud microphysical processes, including aerosol cloud interactions.

The outcome of the study is a modified parameterization of the cloud droplet shape parameter, as commonly used in two-moment bulk microphysical models.

(Answer) We would like to thank Anonymous Referee #2 for taking the time to analyze our work and suggest improvements. In this document we provide detailed responses to the issues raised.

**Major points:**

1. (Comment) Personally, after reading this manuscript (and parts of its precursor, Cecchini et al. 2017) I am still having a hard time to see the point of using the "gamma phase space" for interpretations of physical processes, or even just to compare differences between models and/or measurements. By recognizing the units of the parameters, it is obvious that the physical interpretation of these parameters is far less straightforward or intuitive than looking at the moments and the corresponding change rates - number, mass, and maybe surface or reflectivity if we want to add a third moment (thus constraining all three parameters  $N_0$ ,  $\Lambda$ ,  $\mu$ ). In the end, the authors seem to set aside their previously introduced method, and use the moments or a combination of moments instead, stating it yields additional physical insight. Large parts of the results section provide only short descriptions of what the parameters look like in the plots, while the actual interpretation is done based on the moments. I cannot see which conclusion of the paper would not have been possible by looking at the moments only. Also the outcome of their new parameterization is a function of bulk number and mass (eq. 11). So why should I make an additional effort in future and explicitly interpret the parameters, and why should I call a process rate pseudo force?

1. **(Answer)** Changes from one position to another in the Gamma phase space are directly linked to changes in the DSD, so the trajectories in this space correspond to effect of the different microphysical processes acting in the cloud (cloud top, in this case).

We believe both approaches, i.e. interpretation of moments or Gamma parameters,

**ACPD**
are incomplete when studied alone. Note that when 3 moments are known, while you can understand a lot about the bulk nature of the droplet population, there is a lack of information about the overall DSD shape and appearance. When you have the Gamma parameters, you know how the DSD will look like but have a hard time converting to bulk quantities without the use of computations. Each approach has its advantages and disadvantages depending on the application. For instance, DSD width (conventionally associated to  $\mu$  in the Gamma case) is very important for collision-coalescence parameterization. On the other hand, precipitation retrievals by remote sensing mostly care about bulk quantities. As such, it is clear that both approaches complement one another by providing additional information about the nature of the droplet population.

While we agree that the values of the parameters have a non-trivial physical interpretation, we still have to study them as-is because a lot of applications rely on them. More important than their actual value, at least in our case, are their relative variations. For example, the isolated information of  $\Lambda = 1 \ \mu m^{-1}$  might not tell much. But if this value changes to  $\Lambda = 3 \ \mu m^{-1}$ , then some interpretations are possible. The bigger- $\Lambda$ -DSD is likely associated to higher number concentrations in the right tale of the DSD, because  $\Lambda$  controls the slope of the exponential part of the DSD, which dominates when  $D \to \infty$ . So the increase in  $\Lambda$  is a measure of the effect of growing processes on the DSD. This kind of analysis is likely useful for DSD physics theory, which must be brought to "reality" by the analysis of the corresponding moments.

What the Gamma phase space brings is a simple and direct way to analyze such theoretical DSD patterns – when they are widening/narrowing, when some averaged D is growing/shrinking, etc. What the Gamma phase space does not bring is a simple and direct way to study DSD moments – even though this can be done by coloring the trajectories according to any DSD variable that can be obtained from the Gamma parameters.

The manuscript was edited in order to highlight the complementary nature of both

**ACPD**
approaches – the analysis of the gamma parameters and the DSD moments.

See next comment for our answer regarding the proposed new parameterization.

2. (Comment) A new parameterization of  $\mu$  is proposed, and since the manuscript advertises the gamma phase space I was expecting something like  $\mu$  being a function of other parameters - e.g., for raindrops it is common practice that  $\mu = f(\Lambda)$ . This is not even mentioned, instead the authors continue to rely on moments of the DSD. While I am not saying it is a bad idea, I cannot see how the gamma phase space has contributed here. Unfortunately, there is no indication of whether the new parameterization would be applicable to any other situations.

2. (Answer) To properly predict DSD moments, model parameterizations should emulate the underlying physics of the problem, which is seen as trajectories in the Gamma phase space, in our case. Therefore, one given parameterization should be able to produce similar trajectories to the benchmark reference chosen – be it a bin model or observations. That is the contribution of the gamma phase space here, independently of the means employed to adjust the trajectory of the bulk parameterization. Even though the new parameterization relies on DSD moments, it does not mean the Gamma phase space wasn't used. In fact, what we did was to benefit from noting that the addition of  $1/q_c$  in Eq. (11) produced a gamma phase space trajectory much closer to the bin case as compared to the original parameterization. Therefore, the new parameterization better reproduces the DSD physics relying on the same  $q_c$  and  $N_d$  values. Note that while the Gamma phase space is not directly present in Eq. (11), it was essential to the development of the new parameterization. In this specific case, the gamma space was employed to understand the former  $\mu$  parameterization, to test different hypothesis and to confirm the best  $\mu$  adjustment.

It is all about the choice of the reference system. We agree that we could have validated the cloud-top path in a bulk phase space instead, and then emulate it to conceive a new parameterization for  $\mu$ . However, as stated in our response to major point 1, our point
is that the gamma phase space approach is a more useful way to analyze the evolution of a DSD, even if it is not straightforward.

Of course it may be more convenient/accurate ways to reproduce observed gamma phase space characteristics in bulk models, and we continue to work into it. One of them could be the definition of additional relations between the gamma parameters (e.g. in the form of the mentioned  $\mu = f(\Lambda)$  as in Zhang et al., 2003), which would imply a slightly large modification to current parameterizations, since we would have to change the method to solve the system of equations for the gamma parameters. It shouldn't be much difficult, but the objective of this paper is to bring up the gamma phase space utility from a modeling approach -for testing, evaluating and developing the parameterizations- rather than presenting a detailed implementation of those ideas. This study presents an initial modeling insight of the gamma phase space, analogously to the work of Cecchini et al. (2017), and exemplify how current parameterizations can benefit from it. It is already the subject of our current research, we are developing a new parameterization for the gamma parameters based in preferential directions of the pseudo-forces in the gamma space.

The manuscript was modified in order to clarify these aspects.

3. (Comment) Another example is how the manuscript addresses the effect of aerosols on cloud properties. Figure 3 shows the sensitivity of the parameters to aerosol concentrations. The main dependency seems to be represented by the magnitude of  $N_0$  (while it is really challenging to see even qualitative dependencies in the 3D plots).  $N_0$  is proportional to the bulk number (zeroth moment), but at the same time a complicated function of  $\mu$  and  $\Lambda$  (which already span the other two dimensions). So the only effect of substituting bulk number by  $N_0$  is that the analysis becomes more complicated or even meaningless - personally I cannot calculate  $\frac{\Lambda^{\mu+1}}{\Gamma(\mu+1)}$  without using tools. On the other hand, the dependence of bulk number on aerosol concentration is well-established.

3. (Answer) We added projections on the three planes of Fig. 3 in the manuscript,
as reproduced in Fig. 1 here. The projections greatly facilitate the interpretation of the trajectories. Aside from the  $N_0$  difference, we also highlight the differences in the  $\Lambda - \mu$  plane. Note that the curves in this plane can be approximated by straight lines and the differences among them are mainly associated to their angular coefficient. Using the effective diameter  $D_{eff} = \frac{\mu+3}{\Lambda}$  as an example, we observe that the angular coefficients are related to droplet growth with cloud height. The higher the coefficient is (in absolute terms) the faster the droplet growth will be. Given that the cleanest case is associated to the highest coefficient, it is also associated to the fastest growth rates. Therefore, aerosols affect not only droplet number concentrations but also their growth rates throughout the whole warm phase – which is already studied in the literature and is usually justified by the water vapor competition process. The point is that the trajectories provide a more complete view of the aerosol effect by showing the changes in all DSD properties at once – at least under the Gamma limitations.

This discussion was added to the manuscript for completeness.

4. (**Comment**) In the discussions about forces I do not understand why advection is considered one of them. When I imagine to be sitting within a parcel below cloud top that is being transported upward: Why would advection impose any changes on me, while I am moving along with the parcel and my direct environment as well? I am not resting at one level.

4. **(Answer)** In the manuscript, the discussion about the pseudo-forces refers to Fig. 4 and Fig. 5. Since this comment mentions a parcel below cloud-top that is being transported upward, we assumed it refers to the pseudo-forces analysis in Fig. 5. Note that in Fig. 4 we do stay in one constant level.

The no-advection assumption is only applicable in closed systems, such as in adiabatic parcel models. In our single-column simulations, what we actually have is an Eulerian framework, where following a parcel is non-trivial, if not impossible, which is pretty much what happens in the atmosphere. At every time step, in addition to the micro-

**ACPD**
physics processes, the source and/or sink effect of the advection is calculated for each model grid-point, determining a continuous mixing between them. Therefore, when we follow the cloud-top, we are dealing with particles that arrived from inferior layers, as well as with those that were nucleated there. In other words, there is no mass conservation for a model grid-point. Also note that the pseudo-forces represented in Fig. 5 (in the manuscript) correspond to averages for the time-steps the cloud-top remained at the same height.

The explanation above was added to the manuscript for clarity.

5. (**Comment**) The main message I am taking away is that current parameterizations of the cloud droplet shape parameter, oftentimes a function of droplet number, are probably not in a final stage yet and there is room for improvements. This confirms what was recently described by Igel and van den Heever (2017). At the same time, the possibility of using 3-moment bulk schemes to explicitly predict the shape parameter based on the microphysical processes is hardly mentioned in the manuscript.

5. **(Answer)** Indeed, large uncertainties are still associated to the shape parameter characterization in bulk microphysics schemes, in addition to the assumption of a predefined functional relationship for droplets size distributions. That is one of the reasons why, even having its own uncertainties, bin schemes are considered more realistic and are often taken as a reference for improving bulk parameterizations. To notice the fact that triple-moments parameterization are already an alternative to overcome this problem, we added the next sentence to line 10, page 2 of the manuscript: "Although triple-moments schemes already allow to determine the three parameters of the gamma function without additional considerations (Milbrandt and Yau, 2005a,b; Szyrmer et al., 2005), they are still too computationally costly for many applications of practical interest, such as operational forecasts or even research activities."

6. (Comment) As part of the discussion within ACPD, I also want to comment on the criticism of Reviewer 1. The paper of Xue et al. (2017) is cited in order to establish that

**ACPD**
in practice, bin microphysical models may not be useful because they yield a spread in the results that is comparable to bulk models. I am going to explain why I cannot agree with the reviewer's opinion who claims that we cannot trust the bin model used here: Xue et al. (2017) present a model intercomparison of three bin models which simulate a squall line in an idealized setup. They find considerable differences which are solely attributable to the microphysical processes and their representation. However, the very point of the paper is that even bin models – whose primary advantage is a free evolution of the particle size distribution – are still relying on and suffering from a number of assumptions related to ice microphysics such as particle densities, shapes, conversion thresholds, treatment of liquid fractions, etc. On the other hand, liquid-only microphysics are way less ambiguous, even though we can think of slightly different relations for fall velocities, coalescence efficiencies and other details. Therefore the heavy criticism seems inappropriate.

6. **(Answer)** As we answered for Anonymous Referee #1, we also agree that the Xue et al. (2017) study should not be used to discourage the working with the TAU model. We understand the criticism from Anonymous Referee #1 because bin models may indeed produce high uncertainties even if they have better representation of physical processes. One example of the sources of those uncertainties in bin schemes is the spurious broadening that occurs due to numerical diffusion in physical space during condensational growth/evaporation (Morrison et al., 2018, JAS). But the point of using the bin model as reference in our study is because it is supposed to better represent the warm phase microphysical processes (namely nucleation, condensation and collision-coalescence). The 1D bin model specifically is possibly one of the best tools for such analysis because it isolates the microphysical processes from more complex and non-linear dynamical interactions. If we can emulate bin results in a bulk scheme, it should be a significant step forward – and that is one of the ideas behind the new parameterization introduced.

7. (Comment) While I regret I could not extract the essence of the manuscript regard-
ing the gamma phase space, I am going to provide a number of specific suggestions for improvements to the manuscript. Generally, I find a lot of vague sentences and I wished there were more information and more specific sentences in all parts of the manuscript.

7. **(Answer)** We appreciate the effort from Anonymous Referee #2 to suggest improvements for our manuscript. We are sure the manuscript is now more explanatory, clear and consistent.

**Specific points:**

1. **(Comment)** Page 1 Lines 22,23: What are practical applications? "Generally employed" is very vague or even wrong. Also references might help.

1. **(Answer)** The text was modified to "Although bin schemes are more accurate and flexible (Berry and Reinhardt, 1974; Enukashvily, 1980; Tzivion et al., 1987), their high computational cost makes them less useful for operational applications or for research activities that do not focus on the effects of microphysics processes. For most of those applications, bulk schemes are more frequently employed (Lin et al., 1983; Ferrier, 1994; Thompson et al., 2008; Morrison et al., 2009)"

2. (Comment) Line 5: N has units of  $cm^{-3}$  only when integrated over a finite size interval. There seem to be inconsistencies with units also in other places, see below.

2. **(Answer)** In page 2, line 5, we are not actually talking about N (the zeroth moment of the DSD), but of N(D), which, in fact, has units of  $cm^{-3}\mu m^{-1}$  or, in other words, number of droplets with diameter D per  $cm^3$  of air.

3. (Comment) Lines 6-10: vague formulation, what are "enough" moments. The impact of  $\mu$  on cloud water path and condensation rates are described, but which  $\mu$  are we talking about – cloud, rain, ice? Three-moment schemes have been introduced more than 10 years ago and it would be appropriate to mention at this point.

3. (Answer) The intended meaning for "enough moments" was "a number of moments

**ACPD**
enough to obtain a fully determined system of equations". For clarity, the paragraph was edited: "To solve for the three parameters of the gamma function, three moments would be necessary. However, most bulk microphysical parameterizations – single-or double-moment schemes – do not predict enough moments of the DSD to properly describe their variability. As a closure, the  $\mu$  parameter of the gamma DSD is commonly fixed or evaluated (Grabowski, 1998; Rotstayn and Liu, 2003; Morrison and Grabowski, 2007)." This modification also specify that we are referring to the  $\mu$  parameter of droplet size distributions. As mentioned earlier in this document, a sentence was introduced to note the existence of triple-moment schemes.

4. **(Comment)** Lines 13-18: Very vague: What is very useful about it, what are the specific advantages? What are the new opportunities?

4. **(Answer)** As this information is used just to introduce the subject of the research described in the submitted manuscript, we don't believe it would be appropriate to specify many details about the paper of Cecchini et al. (2017). We would prefer to keep this paragraph in its current form, suggesting the reader to find more detailed information in the aforementioned publication.

5. (**Comment**) Page 3 Line 8: what is the sounding date, also there is a lack of information about the AC09 flight. It is cited further below but still it would be nice to have a quick overview including date and the cloud we are looking at.

5. **(Answer)** To include the suggested information, we updated the corresponding paragraph: "As initial conditions, vertical profiles of potential temperature and water vapor mixing ratio from an in situ atmospheric sounding were provided (Fig. 1a). We used the 12Z sounding, on September 11, 2014, from Boa Vista-RR, Brazil, for coherence with the atmospheric conditions where the data of the AC09 flight were collected (Wendisch et al., 2016), intending to use those measurements for comparisons here. This flight was performed by the High Altitude and Long Range Research Aircraft (HALO) on the same date of the aforementioned sounding, as part of the ACRIDICON-CHUVA **ACPD**
campaign (Machado et al., 2014). It sampled the top of growing convective cumulus, starting close to the local noon, over remote regions of the Amazon, where there is relatively homogeneous conditions, due to the characteristics of the surface, and low aerosol concentrations."

6. (**Comment**) Page 4 Line 2: What does the prognostic variable represent? What are the initialized aerosol properties in the model?

6. **(Answer)** The requested information was added to the text: "Aerosols are represented by a single prognostic variable, its bulk number concentration, that was initialized as 800 cm-3. It is assumed to have a log-normal distribution, with a median radius of 0.05  $\mu$ m and a geometric standard deviation of 1.5. The hygroscopicity of the aerosols was considered as 0.1, according to previous characterizations of the aerosol over the Amazon (Gunthe et al. 2009, Martin et al., 2010, Pöhlker et al., 2016)."

7. (Comment) Lines 5-10: The statements about ice properties seem to be irrelevant for this study. Since  $\mu$  is a central topic here, what are the underlying observations/ cloud types/ reference other than Thompson?

7. **(Answer)** We intended to describe main aspects of both parameterizations, that is why we mentioned the species they include and the size distribution function used for them. However, we agree that this information is irrelevant, so we deleted it.

At this point, we just explained the scheme of Thompson et al. (2008), so we believe the introduction section would be more appropriate to include some content about observations of  $\mu$ . According to that idea, we added a reference to Miles et al. (2000), who summarized several values of  $\mu$  previously reported in the literature.

8. (Comment) Line 18: It seems worth noting that the exponential is a gamma distribution with  $\mu = 0$ .

8. **(Answer)** We now included this statement in the paragraph where the gamma distribution is presented, in the introduction.

ACPD
9. (**Comment**) Line 26: The morrison scheme uses SI units. Also it does not use mass densitites, but mixing ratios.

9. (Answer) Yes, we are aware that the Morrison scheme uses mixing ratios and SI units instead, but it does not affect the expressions for  $\mu$ ,  $\Lambda$  and  $N_0$  presented. They are equivalent in both schemes.

10. (Comment) Lines 29 and following: I do not understand: What is different in the approach of Morrison to estimate the parameters? Line 22 states that both schemes use the same expressions. I am also curious whether potential differences between the schemes in terms of units are considered correctly. Trying to understand: The Morrison scheme is not used to calculate any process rates, but only to diagnose DSD parameters? Are the expected differences due to the parameterization of  $\mu$  or anything else? If so, why not simply replace the parameterization of  $\mu$  within the Thompson scheme with the one used in the Morrison scheme? But in contrast to the Morrison scheme, the Thompson microphysics does calculate its own process rates? How can the comparison be fair when Morrison gets the moment input from TAU, but Thompson predicts the moments on its own? What are "the uncertainties introduced by the procedure..." – only the parameterization of  $\mu$ ?

10. **(Answer)** As explained in the manuscript, section 2.1, the only difference between the way these two bulk schemes estimate the gamma parameters lies on the calculation of  $\mu$ , since they use the same expressions for  $\Lambda$  and  $N_0$ . To address the concern of Anonymous Referee #2 about the units in the scheme of Morrison, let's consider the expression for  $\mu$  originally employed on it:

$$\mu_M = (0.0005714N_M\rho_M \times 10^{-6} + 0.2714)^{-2} - 1 \tag{1}$$

where  $N_M$  is the droplet number concentration (kg-1) and  $\rho_M$  is the density of the air (kg.m-3). Note that  $N_M \rho_M \times 10^{-6}$  becomes  $N_d$  (cm-3), and Eq. 1 here equals Eq. 4 of the manuscript. Then, using the same  $N_d$  (cm-3), we can obtain consistent values
of  $\boldsymbol{\mu}$  for each scheme.

However, considering the schemes in its entirety, there are many more differences among them than just the calculation of  $\mu$ . If we want to compare its gamma trajectories, it is useful to avoid the influence of other aspects of the schemes. That is what we do when we calculate the gamma parameters of the Morrison's scheme and recalculate the ones of the Thompson's from the moments predicted by TAU. Firstly, we run TAU and Thompson and compare the trajectories, they are obviously different, then we say "ok, let's test whether the differences in the gamma trajectories have a large influence of other features aside from the way they estimate  $\mu$ , let's use the same base moments. And the answer is "No, even if somehow they could obtain the same moments, the estimation of the gamma parameters will be incorrect". Of course, obtaining different gamma parameters will influence conversion processes rates and mass and number concentration predictions.

We don't compare Morrison's scheme getting the moment input from TAU, with Thompson's scheme predicting the moments on its own, that wouldn't be fair, indeed. We compare Thompson, predicting the moments on its own with the results of TAU, and then Thompson and Morrison, getting the moments from TAU, with TAU.

11. (**Comment**) Page 5 Line 8: Please explain "process intensities", or otherwise it would be helpful to stick to established wording.

11. (Answer) "Process intensities" was edited to "rates of those processes".

12. **(Comment)** Lines 10-14: What makes the phase space a projection? Is is something different from the 3d space that is used here?

12. **(Answer)** We substituted the word "projection" by "representation", to avoid ambiguities.

13. (Comment) Line 20: Please explain the restriction: Are there considerable amounts of drizzle/rain present which is just cut off from the DSD? Is the intention

ACPD
to avoid having a second mode in the DSD, or are there other reasons? 50 micron appears pretty small indeed. Even though most of the bulk number will be contained at sizes below this threshold, considerable fractions of mass can be contained in the tail larger than that. This may be also a reason for the big values of diagnosed shape parameters.

13. **(Answer)** Please see response to major point 3 of Anonymous Referee #1. The 50 microns restriction intend to avoid raindrops, to be consistent with the analysis of Cecchini et al. (2017). It is a commonly used threshold in the literature to distinguish cloud droplets from raindrops. Nevertheless, there is no significant quantity of drops beyond that size interval in the simulated cloud-top (information added to the manuscript).

14. (**Comment**) Lines 28 - 31: The "bulk phase space" is another example when I feel that new content is created by new wording only. The authors interpret the moments of the DSD in order to get an idea about the physical processes, which has been done by the community for decades. Since the authors also see the need to do so, I am concluding that the "gamma phase space" as such is limited in being useful to interpret the physics.

14. (Answer) We call the  $N_d - D_{eff}$  phase space as "bulk phase space" for simplicity. Its a phase space defined by bulk properties of the DSD, we are not saying that it is new content. Actually, because understanding physical processes in terms of DSD moments is a more common approach, we use it as a complement of the gamma phase space analysis, which is not so common. As we commented earlier in this document, we believe both approaches complement together and that, despite being more abstract, the information the gamma phase space provides can't be obtained from a single bulk phase space.

15. (**Comment**) Page 6: Line 9: There seems to be a hint that the real cloud contained ice, but the model does not? What does it mean that the cloud was limited to lower heights in the model?
15. **(Answer)** Indeed, since we are using a warm phase bin microphysics scheme as main tool, there is no way to simulate ice processes, any simulated variable above the freezing level would make no sense. Thus, the simulation didn't reach the highest levels of the troposphere, while real cumulus did. To facilitate the understanding, we edited this statement in the manuscript to: "The simulation did not reach the highest levels sampled in the observations because it includes only the warm-phase processes".

16. (Comment) Line 10:  $\mu$  is commonly referred to as the width of the DSD – isn't it a sufficient criterion for a broadening DSD to find a decrease in  $\mu$ ? How are  $N_0$  and  $\Lambda$  important in interpreting the broadening? Could we also think of opposite tendencies for  $N_0/\Lambda$  and still call it broadening?

16. **(Answer)** The habitual association between the DSD width and  $\mu$  may come from the relative dispersion ( $\epsilon$ ) concept:

$$\epsilon = \frac{\sigma}{D_m} = \frac{1}{\sqrt{\mu + 1}} \tag{2}$$

where  $\sigma$  is the standard deviation and  $D_m$  is the mean diameter of the DSD. However, decreasing  $\mu$  may not be associated to such an intuitive DSD broadening, even if the relative dispersion increases. Figure 2 here illustrates that when  $\mu$  decreases and the other parameters remain constant, the DSD actually shrinks.

To address this question, let's analyze some characteristics of the gamma DSD, given by Eq. 1 in the manuscript. Taking the first derivative of Eq. 1,

$$\frac{dN}{dD} = N_0 D^{\mu-1} e^{-\Lambda D} (\mu - D\Lambda)$$
(3)

it can be determined that the maximum of the function is located at  $D_{max} = \mu/\Lambda$  and the value of this maximum depends on the gamma parameters according to:
$$N_{max} = N_0 \left(\frac{\mu}{e\Lambda}\right)^{\mu} \tag{4}$$

Now, if we choose two points located at both sides of  $D_{max}$ ,  $D_1$  and  $D_2$ , such that:

$$N(D_1) = N(D_2) = \frac{1}{n} N_{max}$$
(5)

where  $n \in \mathbb{N}$ , we can find the values of  $D_1$  and  $D_2$  by solving:

$$D^{\mu}e^{-\Lambda D} = \frac{1}{n} \left(\frac{\mu}{\Lambda}\right)^{\mu} e^{-\mu} \tag{6}$$

Equation 6 has real solutions in the form of Lambert function branches 0 ( $W_0$ ) and -1 ( $W_{-1}$ ):

$$D_1 = -\frac{\mu}{\Lambda} W_0 \left( -\frac{1}{n} e^{-1} \right) \tag{7}$$

$$D_2 = -\frac{\mu}{\Lambda} W_{-1} \left( -\frac{1}{n} e^{-1} \right) \tag{8}$$

where  $W_0$  and  $W_{-1}$  are negative in the interval  $(-e^{-1}; 0)$ , so  $D_1$  and  $D_2$  are positive.

From the previous analyses, we can see that the relation  $\mu/\Lambda$  determines  $D_{max}$  and  $N_{max}$ , as well as the difference  $D_1-D_2$ . It means that decreasing  $\mu$ , actually decreases the location of the maximum, its magnitude, and the distance between the ascending and descending branches of the gamma function. The interpretation depends on the definition of "broadening". While decreasing  $\mu$  causes a relative broadening (expressed by the increase of  $\epsilon$ ), for an absolute broadening of the DSD, the decrease in  $\mu$  must

**ACPD**
go along with a decrease in  $\Lambda$ , as illustrated in Fig. 2 here. An increment in the value of  $N_0$  will act only as a scale factor, increasing the bulk number concentration of the DSD.

The previous discussion was added to the text of the manuscript for completeness.

17. (**Comment**) Figures: Is log() referring to the natural logarithm? In particular to interpret the numerical values of  $\mu$ ,  $log_{10}()$  scales will be much more intuitive. At the same time I wonder if the values considerably larger than, say 20-30, are pointing to problems in the diagnosis of  $\mu$ . The 3D plots provide hardly any usable information, even qualitative judgements are difficult. It would make sense to show the data under discussion in a 2D plane or some other kind of restructuring.

17. **(Answer)** Yes, log() referred to the natural logarithm in the figures. The intention was to be coherent with the analysis of Cecchini et al. (2017). However, there was a misunderstanding partially motivated by the terminology used. We noted that Cecchini et al. (2017) actually used  $log_{10}()$ , so we updated the corresponding figures in the manuscript. Figure 3 here is a reproduction of Fig. 2 in the manuscript, now log() means  $log_{10}()$ . There we can see that the gamma phase space paths in the simulation and in the observation are actually in a better agreement than what was illustrated by the original figure.

We discussed the concern about the large values of  $\mu$  in the answer to major point 4 of Anonymous Referee #1. In that document, we show that large values of  $\mu$  corresponds to very incipient DSDs, when freshly activated droplets occupy only smaller-diameter intervals.

In the updated version of the manuscript we added projections to the 3D plots – as in Fig. 3 here–, it allows for interpreting the data both in 3D and 2D simultaneously.

**References**

Berry, E. X. and Reinhardt, R. L.: An analysis of cloud drop growth by collection: Part
I. Double distributions, Journal of the Atmospheric Sciences, 31, 1814–1824, 1974.

Cecchini, M. A., Machado, L. A. T., Wendisch, M., Costa, A., Krämer, M., Andreae, M. O., Afchine, A., Albrecht, R. I., Artaxo, P., Borrmann, S., Fütterer, D., Klimach, T., Mahnke, C., Martin, S. T., Minikin, A., Molleker, S., Pardo, L. H., Pöhlker, C., Pöhlker, M. L., Pöschl, U., Rosenfeld, D., and Weinzierl, B.: Illustration of microphysical processes in Amazonian deep convective clouds in the gamma phase space: introduction and potential applications, Atmospheric Chemistry and Physics, 17, 14 727–14 746, 2017.

Enukashvily, I. M.: A numerical method for integrating the kinetic equation of coalescence and breakup of cloud droplets, Journal of the Atmospheric Sciences, 37, 2521– 2534, 1980.

Ferrier, B. S.: A double-moment multiple-phase four-class bulk ice scheme. Part I: Description, Journal of the Atmospheric Sciences, 51, 249–280, 1994.

Grabowski, W. W.: Toward Cloud Resolving Modeling of Large-Scale Tropical Circulations: A Simple Cloud Microphysics Parameterization, Journal of the Atmospheric Sciences, 55, 3283–3298, 1998.

Gunthe, S. S., King, S. M., Rose, D., Chen, Q., Roldin, P., Farmer, D. K., Jimenez, J. L., Artaxo, P., Andreae, M. O., Martin, S. T., and Pöschl, U.: Cloud condensation nuclei in pristine tropical rainforest air of Amazonia: size-resolved measurements and modeling of atmospheric aerosol composition and CCN activity, Atmospheric Chemistry and Physics, 9, 7551–7575, 2009.

Igel, A. L. and van den Heever, S. C.: The Importance of the Shape of Cloud Droplet Size Distributions in Shallow Cumulus Clouds. Part I: Bin Microphysics Simulations, Journal of the Atmospheric Sciences, 74, 249–258, 2017.

Lin, Y.-L., Farley, R. D., and Orville, H. D.: Bulk parameterization of the snow field in a cloud model, Journal of Climate and Applied Meteorology, 22, 1065–1092, 1983.

Machado, L. A. T., Dias, M. A. F. S., Morales, C., Fisch, G., Vila, D., Albrecht, R.,
Goodman, S. J., Calheiros, A. J. P., Biscaro, T., Kummerow, C., Cohen, J., Fitzjarrald, D., Nascimento, E. L., Sakamoto, M. S., Cunningham, C., Chaboureau, J.-P., Petersen, W. A., Adams, D. K., Baldini, L., Angelis, C. F., Sapucci, L. F., Salio, P., Barbosa, H. M. J., Landulfo, E., Souza, R. A. F., Blakeslee, R. J., Bailey, J., Freitas, S., Lima, W. F. A., and Tokay, A.: The Chuva Project: How Does Convection Vary across Brazil?, Bulletin of the American Meteorological Society, 95, 1365–1380, 2014.

Martin, S. T., Andreae, M. O., Artaxo, P., Baumgardner, D., Chen, Q., Goldstein, A. H., Guenther, A., Heald, C. L., Mayol-Bracero, O. L., McMurry, P. H., Pauliquevis, T., Pöschl, U., Prather, K. A., Roberts, G. C., Saleska, S. R., Dias, M. A. S., Spracklen, D. V., Swietlicki, E., and Trebs, I.: Sources and properties of Amazonian aerosol particles, Reviews of Geophysics, 48, 2010

Milbrandt, J. and Yau, M.: A multimoment bulk microphysics parameterization. Part I: Analysis of the role of the spectral shape parameter, Journal of the atmospheric sciences, 62, 3051–3064, 2005a.

Milbrandt, J. and Yau, M.: A multimoment bulk microphysics parameterization. Part II: A proposed three-moment closure and scheme description, Journal of the atmospheric sciences, 62, 3065–3081, 2005b.

Miles, N. L., Verlinde, J., and Clothiaux, E. E.: Cloud droplet size distributions in lowlevel stratiform clouds, Journal of the atmospheric sciences, 57, 295–311, 2000.

Morrison, H. and Grabowski, W. W.: Comparison of Bulk and Bin Warm-Rain Microphysics Models Using a Kinematic Framework, Journal of the Atmospheric Sciences, 64, 2839–2861, 2007.

Morrison, H., Thompson, G., and Tatarskii, V.: Impact of Cloud Microphysics on the Development of Trailing Stratiform Precipitation in a Simulated Squall Line: Comparison of One- and Two-Moment Schemes, Monthly Weather Review, 137, 991–1007, 2009.
Morrison, H., Witte, M., Bryan, G. H., Harrington, J. Y., and Lebo, Z. J.: Spurious broadening of modeled cloud droplet spectra using bin microphysics in an Eulerian spatial domain, Journal of Atmospheric Sciences, in review, 2018.

Pöhlker, M. L., Pöhlker, C., Ditas, F., Klimach, T., Hrabe de Angelis, I., Araújo, A., Brito, J., Carbone, S., Cheng, Y., Chi, X., Ditz, R., Gunthe, S. S., Kesselmeier, J., Könemann, T., Lavric, J. V., Martin, S. T., Mikhailov, E., Moran-Zuloaga, D., Rose, D., Saturno, J., Su, H., Thalman, R., Walter, D., Wang, J., Wolff, S., Barbosa, H. M. J., Artaxo, P., Andreae, M. O., and Pöschl, U.: Long-term observations of cloud condensation nuclei in the Amazon rain forest – Part 1: Aerosol size distribution, hygroscopicity, and new model parametrizations for CCN prediction, Atmospheric Chemistry and Physics, 16, 15 709–15 740, 2016.

Rotstayn, L. D. and Liu, Y.: Sensitivity of the First Indirect Aerosol Effect to an Increase of Cloud Droplet Spectral Dispersion with Droplet Number Concentration, Journal of Climate, 16, 3476–3481, 2003.

Szyrmer, W., Laroche, S., and Zawadzki, I.: A Microphysical Bulk Formulation Based on Scaling Nor- malization of the Particle Size Distribution. Part I: Description, Journal of the Atmospheric Sciences, 62, 4206–4221, 2005.

Thompson, G., Field, P. R., Rasmussen, R. M., and Hall, W. D.: Explicit Forecasts of Winter Precipitation Using an Improved Bulk Microphysics Scheme. Part II: Implementation of a New Snow Parameteriza- tion, Monthly Weather Review, 136, 5095–5115, 2008.

Tzivion, S., Feingold, G., and Levin, Z.: An efficient numerical solution to the stochastic collection equation, Journal of the Atmospheric Sciences, 44, 3139–3149, 1987.

Wendisch, M., Pöschl, U., Andreae, M. O., Machado, L. A. T., Albrecht, R., Schlager, H., Rosenfeld, D., Martin, S. T., Abdelmonem, A., Afchine, A., Araùjo, A. C., Artaxo, P., Aufmhoff, H., Barbosa, H. M. J., Borrmann, S., Braga, R., Buchholz, B., Cecchini, M. A.,
Costa, A., Curtius, J., Dollner, M., Dorf, M., Dreiling, V., Ebert, V., Ehrlich, A., Ewald, F., Fisch, G., Fix, A., Frank, F., Fütterer, D., Heckl, C., Heidelberg, F., Hüneke, T., Jäkel, E., Järvinen, E., Jurkat, T., Kanter, S., Kästner, U., Kenntner, M., Kesselmeier, J., Klimach, T., Knecht, M., Kohl, R., Kölling, T., Krämer, M., Krüger, M., Krisna, T. C., Lavric, J. V., Longo, K., Mahnke, C., Manzi, A. O., Mayer, B., Mertes, S., Minikin, A., Molleker, S., Münch, S., Nillius, B., Pfeilsticker, K., Pöhlker, C., Roiger, A., Rose, D., Rosenow, D., Sauer, D., Schnaiter, M., Schneider, J., Schulz, C., de Souza, R. A. F., Spanu, A., Stock, P., Vila, D., Voigt, C., Walser, A., Walter, D., Weigel, R., Weinzierl, B., Werner, F., Yamasoe, M. A., Ziereis, H., Zinner, T., and Zöger, M.: The ACRIDICON-CHUVA campaign: Studying tropical deep convective clouds and precipitation over Amazonia using the new German research aircraft HALO, Bulletin of the American Meteorological Society, 2016.

Xue, L., Fan, J., Lebo, Z. J., Wu, W., Morrison, H., Grabowski, W. W., Chu, X., Geresdi, I., North, K., Stenz, R., Gao, Y., Lou, X., Bansemer, A., Heymsfield, A. J., McFarquhar, G. M., and Rasmussen, R. M.: Idealized Simulations of a Squall Line from the MC3E Field Campaign Applying Three Bin Microphysics Schemes: Dynamic and Thermodynamic Structure, Monthly Weather Review, 145, 4789–4812, 2017.

Zhang, G., Vivekanandan, J., Brandes, E. A., Meneghini, R., and Kozu, T.: The shape– slope relation in observed gamma raindrop size distributions: Statistical error or useful information?, Journal of Atmospheric and Oceanic Technology, 20, 1106–11

**ACPD**

**ACPD**

---

## Author Comment (AC3) · 26 Apr 2018

We noted that Cecchini et al. (2017) actually used $log_{10}()$, so we updated the corresponding figures in the manuscript. This does not affect the interpretations of the figures in the manuscript, nor the conclusions extracted from them. However, the gamma phase space paths in the simulation and in the observation are actually in a better agreement than what was illustrated by the original figure (See Fig. 3 of the authors response to Anonymous Referee 2). Here we submit an updated version of Fig. 4 of the authors response to Anonymous Referee 1, where we also added projections of the trajectories on the axis planes.

[Figure]

[Figure]

[Figure]

**Fig. 1.** Updated version of Fig. 4 of the authors response to Anonymous Referee #1

[Figure]